# Organization of a cytoskeletal superstructure in the apical domain of intestinal tuft cells

Jennifer B. Silverman[1] , Evan E. Krystofiak[1,2] , Leah R. Caplan[1] , Ken S. Lau[1] , and Matthew J. Tyska[1]

**Tuft cells are a rare epithelial cell type that play important roles in sensing and responding to luminal antigens. A defining morphological feature of this lineage is the actin-rich apical "tuft," which contains large fingerlike protrusions. However, details of the cytoskeletal ultrastructure underpinning the tuft, the molecules involved in building this structure, or how it supports tuft cell biology remain unclear. In the context of the small intestine, we found that tuft cell protrusions are supported by long-core bundles that consist of F-actin crosslinked in a parallel and polarized configuration; they also contain a tuft cell–specific complement of actin-binding proteins that exhibit regionalized localization along the bundle axis. Remarkably, in the sub-apical cytoplasm, the array of core actin bundles interdigitates and co-aligns with a highly ordered network of microtubules. The resulting cytoskeletal superstructure is well positioned to support subcellular transport and, in turn, the dynamic sensing functions of the tuft cell that are critical for intestinal homeostasis.**

## Introduction

Epithelial tissues are composed of a diverse array of cell types including transporting, sensory, and secretory cells, which work together to support physiological homeostasis. However, not all cell types are equally represented within the epithelium, and certain populations are rare. One such rare cell type is the tuft cell (also known as the brush cell), which is characterized as a chemosensory cell and found in the thymus, gastrointestinal, respiratory, and urogenital tracts (Bezencon et al., 2007; Chang et al., 1986; Hofer and Drenckhahn, 1992; Miller et al., 2018; Saqui-Salces et al., 2011). In the intestinal epithelium, specifically, tuft cells comprise <1% of all cells. Perhaps because of their rarity, our understanding of their basic biological features and how they support epithelial tissue physiology remains understudied.

Tuft cells from diverse tissues perform similar functions dependent on G-protein coupled receptors (GPCRs) sensing ligands specific to their biological niche. In the small intestine, tuft cells sense bacteria, protists, and helminths using a variety of GPCRs including succinate receptor 1 and free fatty acid receptor 3 (Lei et al., 2018; Park et al., 2022). Through the canonical taste receptor pathway, transient receptor potential cation channel subfamily M member 5 causes an influx of sodium that depolarizes the cell (Howitt et al., 2016). Downstream, depolarization causes the secretion of tuft cell effectors including interleukin 25 (IL-25), cysteinyl leukotrienes (CysLTs), prostaglandin D2 (PGD2), and acetylcholine (Ach) (Labed et al., 2018; McGinty et al., 2020;

Oyesola et al., 2021; von Moltke et al., 2016). These effectors trigger a type 2 innate lymphoid cell response which, in the intestine, results in tuft cell hyperplasia and aids in the subsequent clearance of parasites and barrier maintenance (Labed et al., 2018; McGinty et al., 2020; Oyesola et al., 2021; von Moltke et al., 2016). Indeed, while wild-type mice cleared helminth infections after 13 days, mice lacking tuft cells failed to clear the worms (Gerbe et al., 2016), highlighting the important role tuft cells play in maintaining intestinal homeostasis.

Interestingly, tuft cells may be heterogeneous within individual tissues, with evidence for two broad subtypes. Type-1 tuft cells are classified as neuron-like and express choline acetyltransferase, an enzyme required for the synthesis of acetylcholine, whereas type-2 tuft cells are classified as immune-like and express immune-related genes such as pan-immune marker CD45/Ptprc (Haber et al., 2017). Although studies in animals are beginning to reveal how tuft cells contribute to epithelial physiology, the subcellular mechanisms underlying tuft cell function in these diverse contexts remain poorly understood.

Tuft cells were first characterized as "peculiar cells" with a large apical "tuft" of protrusions (Jarvi and Keyrilainen, 1956; Rhodin and Dalhamn, 1956). Superficially, tuft cell protrusions exhibit a fingerlike morphology resembling large microvilli. However, early electron microscopy (EM) revealed that tuft protrusions are supported by large actin-rich structures, which

[1]Department of Cell and Developmental Biology, Vanderbilt University School of Medicine, Nashville, TN, USA;   [2]Vanderbilt Cell Imaging Shared Resource, Vanderbilt University, Nashville, TN, USA.

Correspondence to Matthew J. Tyska: matthew.tyska@vanderbilt.edu.

protrude ~2 µm from the cell surface wrapped in apical membrane and extend many microns deep into the cytoplasm, down to the perinuclear region (Hofer and Drenckhahn, 1992; Sato and Miyoshi, 1997; Trier et al., 1987). These imaging studies also noted microtubules beneath the tuft cell apical surface (Hofer and Drenckhahn, 1996; Luciano and Reale, 1979). Interestingly, EM volume reconstructions of single tuft cells revealed that the subapical cytoplasm contains a large "tubulovesicular structure," although any connection between this structure and the cytoskeletal features alluded to above remains unclear (Hoover et al., 2017). Later, studies using light microscopy and single-cell RNA sequencing (scRNA-seq) reported enrichment of specific cytoskeletal components in tuft cells, including acetylated tubulin (Saqui-Salces et al., 2011), actin-bundling protein fimbrin (Hofer and Drenckhahn, 1992), actin- and Akt-binding protein girdin (Kuga et al., 2017), and actin-binding protein advillin (Bezencon et al., 2008; Esmaeilniakooshkghazi et al., 2020; Ruppert et al., 2020). Notably, advillin has high homology (59%) with villin, a major actin filament bundler in enterocyte microvilli (Marks et al., 1998).

Although previous studies hinted at the unique organization and composition of the tuft cytoskeleton, important questions remain unanswered. How this cytoskeletal specialization supports the physiological functions of tuft cells also remains unclear. Answers to these questions will be essential for understanding how tuft cells are adapted to support homeostasis through sensing and secretion. To address these gaps, we leveraged quantitative light and electron microscopy, in combination with trainable image segmentation, with the goal of building detailed three-dimensional views of the cytoskeletal structures underlying the tuft in intestinal tuft cells. As epithelial cell culture models fail to produce tuft cells, our investigations focused primarily on native tissues. We worked around the problem of tuft cell rarity in this tissue by taking advantage of succinate treatment to increase tuft cell specification (Banerjee et al., 2020); we also used scRNA-seq data sets to find candidate cytoskeletal proteins that drive the tuft cell phenotype. Using these approaches, we defined the organization and packing of actin filaments within tuft cell protrusions and characterized the subcellular distribution of cytoskeletal proteins that exhibit tuft cell–specific enrichment. Our studies identified a striking array of microtubules that exhibit interdigitating co-alignment with core actin bundles, forming a superstructure of parallel cytoskeletal polymers that extend from the apical surface down to the perinuclear region. Extensive decoration of this superstructure with multivesicular bodies and other small membranous organelles suggests that this unique architecture might support vectorial transport along the apical–basolateral axis. These discoveries advance our knowledge of fundamental tuft cell biology and suggest models for how the unique cytoskeleton found in this rare cell type might contribute to sensing and secretion, subcellular activities that are essential for intestinal homeostasis.

## Results

### The tuft cell "tuft" contains ~100 core actin bundles
Herein, we propose and apply a simple nomenclature when referring to specific structures within the tuft, defined here as the full array of microvillus-like protrusions and their supporting core actin bundles (Fig. 1 A). Our hope is that these terms will be adopted by others in the field to facilitate clear communication. To begin defining the structural architecture of the tuft, we used confocal and super-resolution light microscopy to visualize the tuft cytoskeleton. We took advantage of mouse small intestinal tissue sections and whole-mount preparations; these allowed us to visualize single tuft cells in both lateral and en face (top down) views. We first performed high-resolution confocal imaging of tissue sections labeled with fluorescent phalloidin to visualize F-actin. Lateral views of single tuft cells revealed arrays of long, continuous actin bundles ranging from 5 to 12 µm in length (median: 7.0) (Fig. 1, B and C), up to ~10 times longer than the core bundles that support microvilli on neighboring enterocytes (Mooseker and Tilney, 1975). Close inspection of the phalloidin signal along these actin bundles revealed that they extend as a single, continuous structure from the distal end of the protrusion, down to the perinuclear region (Fig. 1 B and Video 1). Imaging whole-mount sections captured just under the apical surface (Fig. 1 D) enabled us to define the number of actin bundles per cell. Tuft cells contained a median of 101 core bundles (Fig. 1 E) and linescans drawn along the bundle axis show that phalloidin signal tapers toward the rootlet end (Fig. 1 F). Although phalloidin intensity is generally proportional to actin filament density, whether such loss of signal toward the rootlet represents a decrease in filament number, or alternatively, inhibition of phalloidin binding by enrichment of an actin-associated protein remains unclear.

Within the small intestinal epithelium, tuft cells exhibit a fusiform morphology, with a narrow apical surface and wider cell body (Nevalainen, 1977; Sato and Miyoshi, 1997). To determine whether the array of core actin bundles is shaped by the overall morphology of the cell, we measured the cross-sectional area of the cell body relative to the area occupied by bundles, moving down from the apical surface in 1.5-µm increments (Fig. 1 G). Right at the apical surface, bundle area and cell area converge, suggesting that the junctional actomyosin belt might constrain bundle spacing in this plane. However, both bundle area and cell area progressively increased below the apical surface, although the rate of increase was higher for the cell body, indicating that the width of the cell body does not limit bundle spreading deeper in the cell (Fig. 1 H). To better characterize the orientation of individual bundles relative to the apicobasal axis, bundles were segmented via trainable Waikato Environment for Knowledge Analysis (WEKA) segmentation (Arganda-Carreras et al., 2017), thresholds were applied to the generated probability maps to reduce noise (see Materials and methods), and structures were then rendered in 3D (Fig. 1 I). Bundles in these reconstructions demonstrated a median tilt of 81.1°, splaying slightly outward from the apicobasal axis (90°) (Fig. 1 J), again with the narrowest point of constriction right at the apical surface. We also examined the packing organization of bundles within the tuft (Fig. 1 K). Enterocyte microvilli exhibit hexagonal packing, which maximizes the number of protrusions and thus apical membrane holding capacity. In the tuft, measurements of the angle formed by groups of three adjacent protrusions revealed a median of 59.3°, suggestive of hexagonal

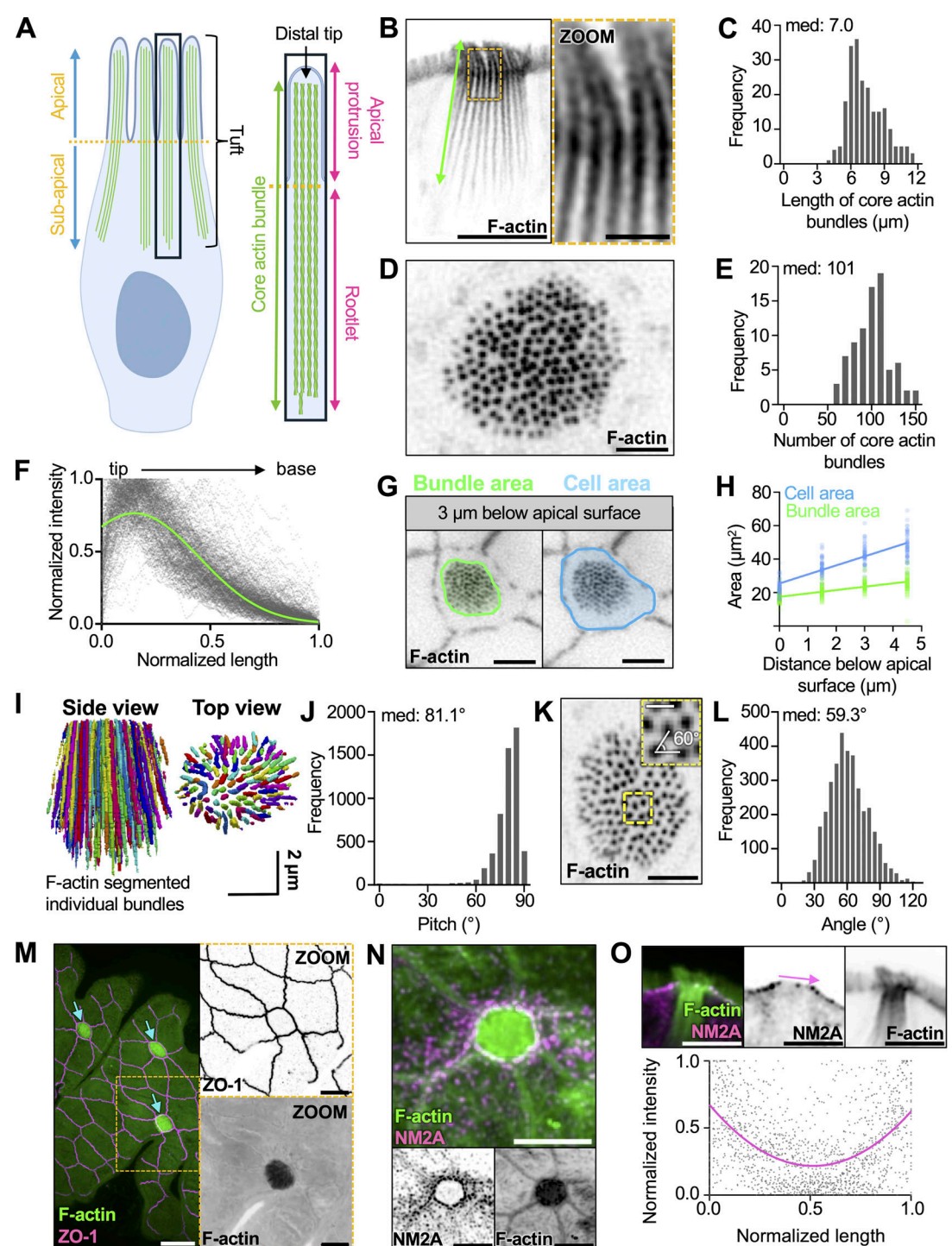

Figure 1. **The tuft cell "tuft" contains over a hundred of core actin bundles. (A)** Cartoon depicting nomenclature for core actin bundles in tuft cells. **(B)** Inverted single channel, single z-slice, Airyscan image of a lateral tissue section of tuft cell. Actin marked with phalloidin (scalebar = 5 µm, zoom = 1 µm). **(C)** Frequency diagram of core bundle length as depicted in B (*n* = 27 tuft cells over 3 mice). **(D)** Max intensity projection (MaxIP) spinning disk confocal (SDC) image using en face whole-mount tissue of a tuft cell captured beneath the apical membrane with phalloidin staining (scalebar = 2 µm). **(E)** Frequency diagram of the number of core actin bundles in tuft cells (*n* = 81 tuft cells over 3 mice). **(F)** Using lateral sections of frozen tissue, linescans of phalloidin intensity were drawn from the tip to the base of core actin bundles. Raw linescan data is shown in gray with a curve fit to data shown in green. (*n* = 218 bundles over 3 mice). **(G)** Single z-slice SDC image of en face whole-mount tissue (scalebar = 5 µm). **(H)** Simple linear regression measuring cell or bundle area (as shown in G) at 1.5, 3, and 4.5 µm beneath the apical surface. Slope bundle area = 2.013. Slope cell area = 5.412 (*n* = 54 tuft cells over 3 mice). **(I)** 3D projection of core actin bundles in a tuft cell using Trainable WEKA segmentation. **(J)** Pitch of individual actin bundles relative to the long axis of the cell (90° = vertical orientation) using WEKA segmented data from I (*n* = 35 tuft cells over 3 mice). **(K)** MaxIP SDC image of en face whole-mount tissue section with showing sub-apical core bundles (scalebar = 2 µm, inset box scalebar = 500 nm). **(L)** Frequency diagram of angle measurements between neighboring bundles as shown in zoom inset K (*n* = 3,453 measurements made in 25 tuft cells over 3 mice). **(M)** MaxIP SDC image of en face whole-mount tissue with ZO-1 (magenta) and actin marked by

phalloidin (green). Arrows point to tuft cells. (scalebar = 5 µm). **(N)** MaxIP SDC image of en face frozen tissue section with NM2A (magenta) and actin marked with phalloidin (green) (scalebar = 5 µm). **(O)** Above: single z-slice image of a lateral frozen tissue section with NM2A (magenta) and actin marked with phalloidin (green) (scalebar = 5 µm). Below: linescans measuring the intensity of NM2A across the apical surface of tuft cells as shown by the pink arrow in the image above (n = 33 tuft cells over 3 mice).

packing, although the spread around this value was large, indicating areas of fluid packing (Fig. 1 L). To gain insight into mechanisms that constrict core actin bundle packing at the apical surface, we examined the actomyosin belt that forms at the level of junctional complexes (Ivanov et al., 2022). Tuft cells have long been identified by their small apical profiles (Sato and Miyoshi, 1997; Trier et al., 1987), and our measurements of en face cell surface area using tight junction marker, zonula occludens 1 (ZO-1) (Fig. 1 M), did reveal that tuft cells exhibit significantly smaller and more circular apical areas than neighboring enterocytes (Fig. S1, A and B). The small radius of the tuft cell apical profile is suggestive of elevated contractility in the junctional actomyosin belt (Ebrahim et al., 2013). Intestinal epithelial cells express three distinct non-muscle myosin-2 isoforms—NM2A, NM2B, and NM2C (McConnell et al., 2011), with NM2A supporting the highest levels of contractility (Kovacs et al., 2003). In the case of tuft cells, en face images showed NM2A enrichment in the junctional actomyosin belt (Fig. 1 N), and lateral views of tissue sections also displayed puncta enriched at the level of junctional contacts (Fig. 1 O); a similar distribution was observed for NM2C (Fig. S1, C and D). To further examine the expression of NM2 genes in tuft cells, we turned to scRNA-seq, which enables cell type–specific assessment of gene expression. Analysis of scRNA-seq data generated from mouse small intestine revealed similar gene expression levels of NM2A between tuft cells and enterocytes (Fig. S1, E and F) and lower gene expression of NM2C in tuft cells (Fig. S1 G). This result was bolstered by immunofluorescence analysis, where NM2A antibody staining intensity was similar between enterocytes and tuft cells, while NM2C intensity was lower in tuft cells (Fig. S1, H and I). These data suggest that the constrained diameter of the tuft cell apical surface could be due to a high NM2A/NM2C ratio.

**Core actin bundles contain ~100 hexagonally packed actin filaments**
We next sought to define the organization of actin filaments within individual core bundles. We used transmission EM (TEM) to collect en face cross-sectional views of the tuft, which enabled us to resolve individual actin filaments in core bundles. To overcome the difficulty of capturing rare tuft cells in ultrathin sections, we increased the number of tuft cells using an established model of succinate-driven tuft cell hyperplasia (Banerjee et al., 2018). With this approach, we found that bundles contain a median of 101.5 filaments (Fig. 2, A and B), in contrast to enterocyte microvilli, which contain only 20–30 (Mooseker and Tilney, 1975). Core bundles exhibited a median diameter of 106.1 nm (Fig. 2 C), whereas whole protrusions demonstrated a median diameter of 152.9 nm (Fig. S2 A). The resulting gap between the core bundle and surrounding membrane was measured at a median of 16.8 nm (Fig. S2 B),

consistent with the dimensions of known membrane-actin linkers, including class 1 myosins (Jontes et al., 1995).

To further examine the arrangement of core bundle actin filaments, we used Fourier analysis to create frequency space maps that highlight dominant spacing features. In image fields containing many bundle cross-sections, Fourier maps were indicative of uniform filament-filament spacing (Fig. 2 D). Fourier patterns were also used to create a filter that was then overlayed with the original image (magenta) to highlight all filaments with uniform spacing (Fig. 2 D). In all bundles examined, most filaments (67–96%) were highlighted, indicating uniform spacing. Because such uniform interfilament spacing is typically associated with hexagonal packing patterns, we next generated Fourier filters to highlight only filaments with hexagonal packing (magenta) (Fig. 2 G). In all bundles analyzed, most actin filaments were hexagonally packed. We then used Trackmate, a FIJI plugin, to identify filament center coordinates with sub-pixel precision (Fig. S2 D). Counting nearest neighbors within a 12-nm radius (approximately twice the width of a single filament), we found that hexagonally packed filaments, indicated by six nearest neighbors, were enriched in the center of the bundle (Fig. S2 C). Additionally, the angle formed by groups of three neighboring filaments formed a median angle of 58.9° (Fig. 2 E), consistent with hexagonal packing. This analysis also revealed that the median center-to-center distance between filaments was 9.2 nm (Fig. 2 F). Finally, we noted that some bundles contained packing anomalies (dislocations) where filaments were entirely missing (yellow arrow in Fig. 2 A), although this did not appear to disrupt the organization of filaments around these voids. To visualize the filaments within core actin bundles along their length, we examined lateral tissue sections via TEM (Fig. 2 H). Using this approach, it was difficult to capture complete views of entire bundles given their length. However, we did observe that bundle thickness and the number of filaments decreased toward the rootlet ends, which is consistent with the decay in the phalloidin signal (Fig. 1 F). The filament packing patterns revealed in these ultrastructural studies hold implications for understanding mechanisms that drive tuft formation, as we discuss in more detail below.

**Filaments in core actin bundles exhibit uniform "barbed-end out" polarity**
Based on our TEM images, core bundle actin filaments appear to be crosslinked tightly, in parallel. However, the orientation of individual filaments within these bundles has not been defined. Filament orientation in this scenario is important because it will constrain models for how these structures contribute to subcellular function. Bundles consisting of filaments with uniform orientation are well-suited for supporting directional motor-driven transport (Berg and Cheney, 2002), whereas mixed orientation bundles are typically associated with contractile

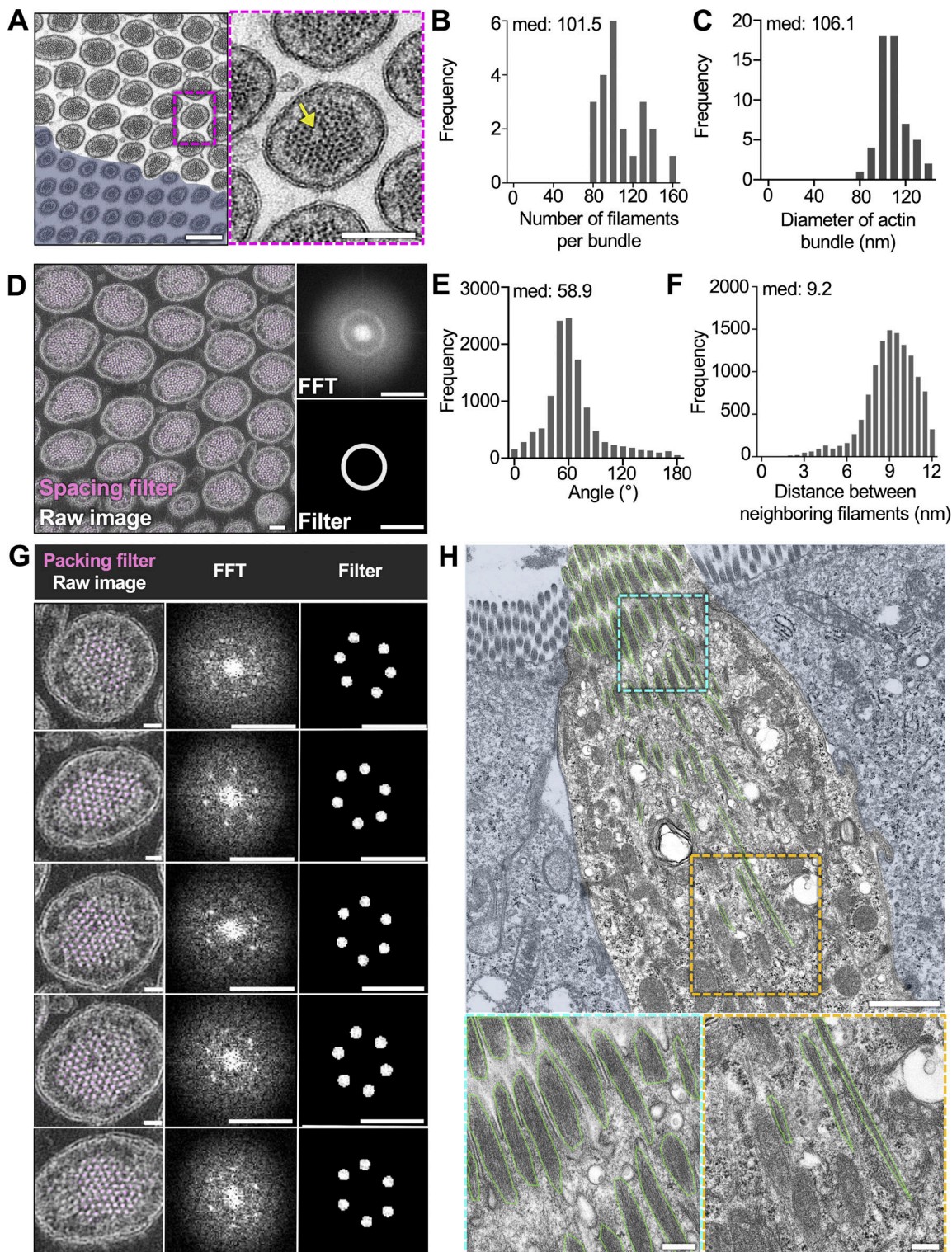

Figure 2. **Core actin bundles contain ~100 hexagonally packed actin filaments. (A)** TEM of ultrathin tissue section depicting tuft cell apical protrusions imaged en face (scalebar = 200 nm) and enterocyte masked in blue, zoom inset on right (scalebar = 100 nm). The yellow arrow points to an instance of actin dislocation. **(B)** Frequency distribution of the number of filaments per bundle ($n$ = 22 bundles over 3 tuft cells. Median values per tuft cell: 108, 102, and 100). **(C)** Frequency distribution showing the diameter of core actin bundles in tuft cell apical protrusions ($n$ = 57 bundles over 3 tuft cells. Median values per tuft cell: 104.4, 108.6, and 103.5 nm). **(D)** TEM of tissue section tuft cell apical protrusions imaged en face. Map of evenly spaced actin filaments (magenta), Fast Fourier transform (FFT) (top right), bandpass filter (bottom right) (scalebar = 100 nm). **(E)** Frequency distribution of the angle between nearest neighbor filaments within a 12-nm radius ($n$ = 22 bundles over 3 tuft cells. Median values per tuft cell: 56.7°, 62.6°, and 64.7°). **(F)** Frequency distribution of the distance between neighboring filaments identified in E ($n$ = 22 bundles over 3 tuft cells. Median values per tuft cell: 8.8, 9, 9.1 nm). **(G)** Map of hexagonally packed filaments on five different protrusions taken from D. FFT (top right), bandpass filter (bottom right) (scalebar raw image = 20 nm, FFT and filter scalebar = 20 nm$^{-1}$). **(H)** TEM of tissue section depicting a mostly lateral view of a tuft cell with enterocytes masked in blue. Core actin bundles are outlined in green (scalebar = 1 μm). Zoom insets on bottom show the base of the rootlets (left) and core bundles near the apical surface (right) (scalebars = 200 nm).

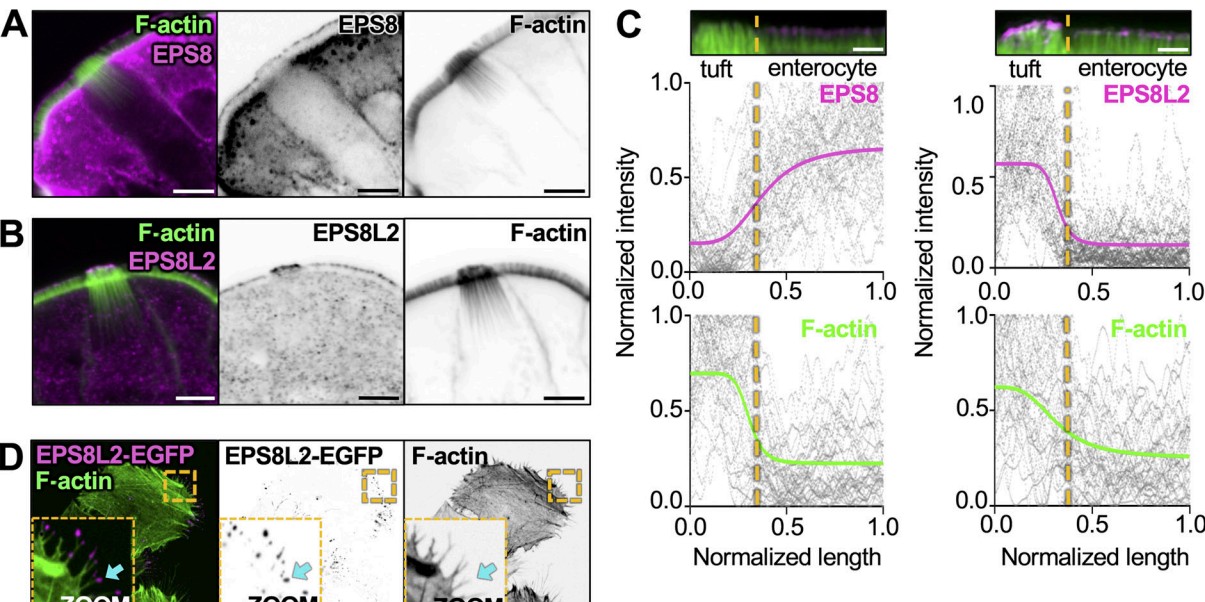

Figure 3. **Filaments in core actin bundles exhibit uniform barbed-end out polarity. (A)** MaxIP of laser scanning confocal image of EPS8 in tuft cells. Actin marked with phalloidin (scalebar = 5 μm). **(B)** MaxIP of laser scanning confocal image of EPS8L2 in tuft cells. Actin marked with phalloidin (scalebar = 5 μm). **(C)** Linescans drawn over a MaxIP image of the apical tuft and neighboring enterocyte microvilli, measuring EPS8 or EPS8L2 intensity in tuft cells and enterocytes, with accompanying phalloidin intensity. Images provided above as reference (scalebar = 2 μm). Raw data for all graphs are shown in gray, line fit (sigmoidal 4PL) in color. The line fit for EPS8 indicates a bottom plateau of 0.15 normalized intensity in tuft cells and a top plateau of 0.66 normalized intensity in enterocytes. The line fit for EPS8L2 indicates a bottom plateau of 0.13 normalized intensity in enterocytes and a top plateau of 0.57 normalized intensity in tuft cells. (n = EPS8: 37 tuft cells over 3 mice; EPS8L2: 33 tuft cells over 3 mice). **(D)** MaxIP SDC image of HeLa cells expressing EPS8L2-EGFP; cyan arrow points to filopodia tip (scalebar = 2 μm).

functions (Tojkander et al., 2012). To determine bundle orientation, we stained mouse small intestine with epidermal growth factor receptor pathway substrate 8 (EPS8), an actin-binding protein that marks barbed-ends of filaments in bundles with uniform orientation (Gaeta et al., 2021). Previous studies established that this factor is highly enriched at the distal tips of microvilli, filopodia, and stereocilia, where the barbed-ends of uniformly oriented filaments reside (Chou et al., 2014; Manor et al., 2011; Postema et al., 2018). Confocal images revealed that tuft cells do exhibit some distal tip targeting of EPS8, although line scan analysis showed significantly lower levels than enterocytes (Fig. 3, A and C). We then stained for other EPS8-like molecules including EPS8-like 2 (EPS8L2), which has a domain organization similar to EPS8 (Offenhauser et al., 2004). Relative to neighboring enterocytes, EPS8L2 is highly enriched in tuft cells where it localizes specifically to the distal tips of apical protrusions (Fig. 3, B and C). Analysis of scRNA-seq data also confirmed the higher level of enrichment for EPS8L2 versus EPS8 in tuft cells (Fig. S3, A–C). Although there is no direct biochemical evidence confirming EPS8L2 as a barbed-end binder, this factor has been shown to target the tips of row 2 stereocilia in hair cells of the inner ear (Furness et al., 2013), which is consistent with barbed-end binding. We also expressed an EGFP-tagged version of EPS8L2 in HeLa cells and observed strong localization to the tips of filopodia (Fig. 3 D), providing additional support for barbed-end binding. Taken together, these data suggest that the tuft cell core actin bundles are composed of filaments organized with a uniform barbed-end out polarity.

**Tuft cell actin-binding proteins exhibit regionalized localization along the core bundle axis**

Cells form higher-order actin structures using a vast array of actin-binding proteins, some of which crosslink and stabilize filaments. For example, stereocilia core bundles contain espin-1, fimbrin/plastin-1, and fascin-2 (Chou et al., 2011; Tilney et al., 1989; Zheng et al., 2000) whereas enterocyte microvilli contain espin, villin, fimbrin, and mitotic spindle positioning protein (MISP) (Bretscher, 1981; Bretscher and Weber, 1979; Loomis et al., 2003; Morales et al., 2022). Recent research on microvilli specifically also indicated that these bundlers occupy distinct "neighborhoods" relative to the ends of the bundle (Lombardo et al., 2024; Morales et al., 2022). In tuft cells, previous studies reported high expression of some actin-binding proteins including fimbrin, advillin, and phosphorylated girdin (Y1798, hereafter referred to as pGirdin) (Esmaeilniakooshkghazi et al., 2020; Hofer and Drenckhahn, 1992; Kuga et al., 2017; Ruppert et al., 2020). By screening scRNA-seq datasets, we also identified additional tuft cell-enriched filament bundling candidates including espin, LIM domain and actin-binding 1 (LIMA1, also known as EPLIN) (Maul et al., 2003), and α-actinin 4 (Honda et al., 1998) (Fig. S3, D–F). To characterize the localization of these proteins in tuft cells, we used super-resolution Zeiss Airyscan microscopy to image immunostained tissue sections. Linescans were drawn along individual core bundles (from distal tip to rootlet), and the signal intensity along that axis was plotted from multiple tuft cells. Advillin displayed high-level enrichment in tuft cells with a signal that was strongest at the

protruding ends of core bundles and decayed toward the rootlets (Fig. 4, A and B). pGirdin signal, in contrast, was restricted to apical protrusions (Fig. 4, C and D). Fimbrin, also found in neighboring enterocyte microvilli, was localized throughout the tuft and like advillin, exhibited the strongest signal in apical protrusions, which then diminished toward the rootlets (Fig. 4, E and F). Espin was also strongly localized to the protruding ends of core actin bundles (Fig. 4, C and D; and Fig. S3 D). Interestingly, LIMA1 was uniquely restricted to the rootlets and maintained a strong signal down to the most proximal ends (Fig. 4, E and F). Finally, α-actinin 4 exhibited staining that was similar to advillin (Fig. S3 F; and Fig. S4, A and B). Thus, espin, LIMA1, and α-actinin 4 are newly identified components of tuft cell core actin bundles, and the regionalization of these factors (Fig. S4 C) likely holds importance for understanding the function and/or mechanical properties of the rootlet versus protruding bundle segments.

Protrusion morphology is maintained by high levels of linker proteins, including class 1 myosins and ERM (ezrin, radixin, and moesin) proteins, which stabilize membrane–actin interactions (Korkmazhan and Dunn, 2022; Morales et al., 2023). From this perspective, we sought to determine if tuft cell protrusions shared similar membrane–actin-linking proteins. Interestingly, scRNA-seq data showed a strong enrichment of myosin-1b in tuft cells compared with enterocytes (Fig. S3 G). As one of eight class 1 myosins expressed in mice, myosin-1b is a tension-sensitive motor implicated in vesicle secretion (Laakso et al., 2008). Immunostaining revealed that myosin-1b is strongly expressed in tuft cells, where it localizes to the protruding ends of core actin bundles (Fig. 5, A and B). Ezrin was also detected in tuft cells (Fig. S3 H) and its expression was similarly restricted to the apical tuft (Fig. 5, C and D). Ezrin is autoinhibited, but it can be activated through the activity of Ste20-like-kinase/ Lymphocyte-oriented 39 kinases (also known as SLK/LOK) (Belkina et al., 2009). Interestingly, whereas total Ezrin levels were similar between tuft cells and enterocytes, tuft cells had significantly higher levels of pEzrin in apical protrusions (Fig. 5, E–G), suggesting that higher levels of membrane–cytoskeleton adhesion may be needed to stabilize the larger dimensions of protrusions in the tuft. Collectively, these immunostaining results highlight a unique complement of actin-associated factors that likely cooperate to drive core bundle assembly and organization within the tuft (Fig. S4 C).

### Core actin bundles co-align and interdigitate with acetylated microtubules

Previous transcriptomic analysis confirmed an enrichment of tubulin in tuft cells (Bezencon et al., 2008), and TEM imaging of ultrathin sections also revealed microtubules between core actin bundles (Luciano and Reale, 1979; Trier et al., 1987). Confocal microscopy of tissue stained for tubulin or acetylated tubulin (Ac-tubulin), a posttranslationally modified version of tubulin associated with long-lived microtubules (Eshun-Wilson et al., 2019), also confirmed the presence of long microtubules in tuft cells (Choi et al., 2015; Hofer and Drenckhahn, 1996; Saqui-Salces et al., 2011). Yet how microtubules in the tuft are arranged relative to the long core actin bundles remains unclear. To address this, we stained for Ac-tubulin in frozen tissue sections and

imaged these samples using confocal microscopy. Consistent with previous work, we observed that microtubules enriched in Ac-tubulin were oriented parallel to the apical–basal axis of tuft cells, but did not enter apical protrusions (Fig. 6 A). This microtubule network also extended several microns past the rootlet ends of core actin bundles, although the overall length of both cytoskeletal networks was comparable (Fig. 6 B). Microtubules are polarized polymers, and in epithelial cells (e.g., enterocytes), they generally extend their minus ends toward the apical surface (Fath et al., 1994; Gilbert et al., 1991). However, when we stained for dynein heavy chain to mark microtubule minus ends, we found minimal dynein enrichment at the apical surface of tuft cells relative to neighboring enterocytes (Fig. 6, C and D), although we noted some dynein labeling towards the perinuclear compartment and basolateral region (Fig. 6 E). Staining for CAMSAP2 and CAMSAP3, both of which bind to minus ends of microtubules (Tanaka et al., 2012), showed a similar pattern characterized by a decreased signal in the sub-apical region of the tuft cell and a lack of defined labeling (Fig. S5, A and B). Interestingly, KIF1C, a microtubule plus end-directed motor (Siddiqui et al., 2022), was enriched in tuft cells and found throughout the cell body, with slightly more labeling of the sub-apical compartment (Fig. S5 C). Collectively, these data indicate that microtubules in the tuft exhibit an organization that is significantly different from the well-characterized "minus end up" network assembled by neighboring enterocytes.

Similar to our analysis of core actin bundles, we used Trainable WEKA segmentation to generate a 3D reconstruction of the Ac-tubulin signal in tuft cells (Fig. 6 F). The overall positioning, organization, and tilt of the microtubules were remarkably similar to core actin bundles (microtubule median: 79.9°; F-actin median: 81.1°) (Fig. 6 I), suggesting the possibility of physical interactions between these two cytoskeletal networks. The strong co-alignment of actin bundles and Ac-tubulin signal was also apparent when superimposing their reconstructions (Fig. 6 G). To better visualize the alignment of these networks, we generated a separate map showing actin bundle segments that were overlapping or immediately adjacent to microtubules (see interaction map, Fig. 6 H). It is important to note that our 3D maps were drawn with conservative thresholding parameters (see Materials and methods), so the full scale of microtubule–actin contact may be underestimated. Despite this, our analysis revealed extended segments of contact ranging up to 4 µm in length (median: 1.0 µm) (Fig. 6 J). Moreover, within each tuft cell, a large percentage (median: 37.8%) of the core actin bundles are marked as interacting with the microtubules using this approach (Fig. 6 K). Taken together, these findings reveal that tuft cell core actin bundles co-align and interdigitate with an array of stable microtubules, forming a cytoskeletal superstructure that extends from the apical surface, down through the sub-apical cytoplasm, to the perinuclear region.

### Membranous organelles associate with cytoskeletal polymers in the tuft

The highly ordered, interdigitated network of acetylated microtubules and core actin bundles within the sub-apical cytoplasm is uniquely positioned and organized to support motor-

Figure 4. **Tuft cell actin-binding proteins exhibit regionalized localization along the core bundle axis. (A, C, E, G, and I)** MaxIP Airyscan image of lateral frozen tissue section and immunostained for actin-binding proteins, (A) advillin, (B) pGirdin, (C) fimbrin, (D) Espin, and (E) LIMA1 in addition to actin marked with phalloidin (scalebar = 5 µm). **(B, D, F, H, and J)** Graph of linescans drawn from MaxIP images from apical tip to the base of core bundle, measuring the intensity of actin-binding proteins. Raw values depicted in gray, line fit (lognormal) in magenta; the yellow line indicates the peak of line fits (*n* = advillin, 29 tuft cells; pGirdin, 32 tuft cells; fimbrin, 30 tuft cells; espin, 33 tuft cells; LIMA1, 32 tuft cells, over 3 mice).

driven trafficking of cargoes between the perinuclear region and apical tuft. Consistent with this proposal, TEM revealed numerous small vesicles and membranous organelles in the submicron range associated with actin bundles and co-aligned microtubules (Fig. 7 A). These include both electron-lucent vesicles (green arrow) as well as potential multivesicular bodies (MVBs, blue arrow). We also observed vesicles that appeared

compressed by the surrounding core bundles, implying direct physical association (see green arrow). Interestingly, TEM images showing cross-sections through apical protrusions revealed an abundance of extracellular vesicles (EVs) situated between apical protrusions. These EVs were similar in dimensions (44 ± 23 nm in diameter) and appearance to the smaller intraluminal vesicles located in multivesicular bodies (45 ± 14 nm in diameter)

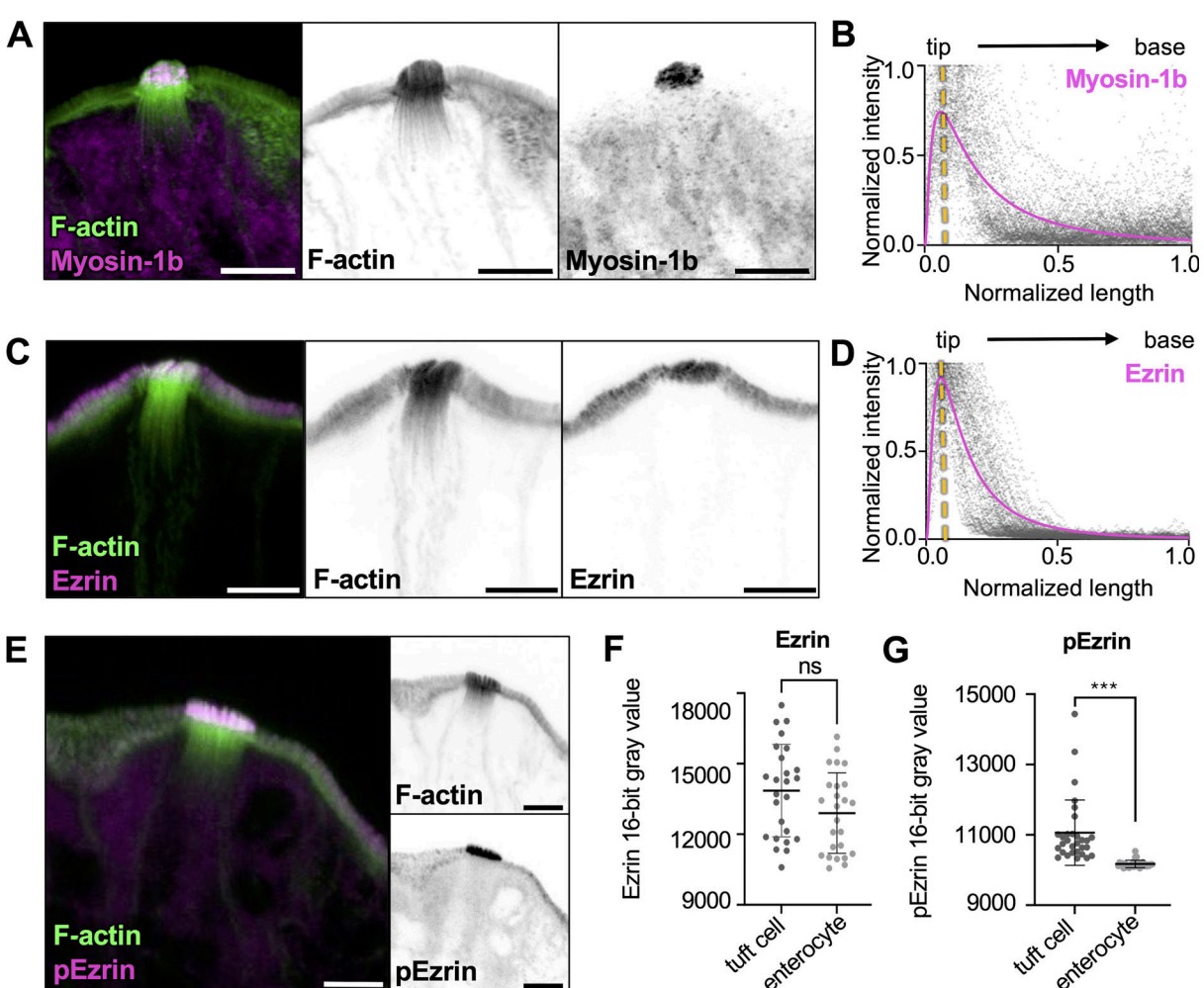

**Figure 5. Core actin bundles show regionalization of tuft cell enriched membrane–actin-linking proteins. (A, C, and E)** MaxIP Airyscan image of the lateral frozen tissue section and immunostained for actin-binding proteins and actin marked with phalloidin (scalebar = 5 μm). **(B and D)** Graph of linescans drawn from MaxIP images from apical tip to the base of core bundle, measuring intensity of actin binding proteins. Raw values are depicted in gray, line fit (lognormal) in magenta; yellow line indicates the peak of line fits ($n$ = myosin-1b, 30 tuft cells; Ezrin, 27 tuft cells, over 3 mice). **(F)** Mean Ezrin intensity in tuft cells versus neighboring enterocytes, using a sum intensity projection, unpaired two-sided $t$ test, P = 0.0723, error bars denote mean ± SD ($n$ = 25 tuft cells over 3 mice). **(G)** Mean pEzrin intensity in tuft cells versus neighboring enterocytes, unpaired two-sided $t$ test, P < 0.001 ($n$ = 30 tuft cells over 3 mice). Error bars denote mean ± SD.

noted in the sub-apical cytoplasm (Fig. 7 B, cyan), suggesting that they might be released by tuft cells into the extracellular space. To examine the possibility that cytoskeletal polymers in the tuft support trafficking of organelles to or from the apical surface, we used light microscopy to examine the distribution of potential membrane protein cargoes. Here, we took advantage of mice expressing endogenously tagged cadherin-related family member 5 (CDHR5), a single-spanning transmembrane protein that resides in apical protrusions in both enterocytes (Crawley et al., 2014) and tuft cells. Imaging revealed a strong CDHR5 signal at the distal tips of protrusions as expected and in large vesicles that were closely associated with Ac-tubulin–containing microtubules in the sub-apical and perinuclear regions (Fig. 7 C). These results suggest that the cytoskeletal superstructure supporting the tuft could play a role in transporting membranous cargoes to and/or from the apical surface.

## Tuft cells exhibit higher levels of trafficking machinery relative to enterocytes

The dense collection of vesicles in the sub-apical cytoplasm and the presence of EVs outside the tuft suggest that tuft cells engage in robust membrane trafficking activities. We therefore stained tuft cells for markers of endo- and exocytosis. Remarkably, tuft cells exhibit robust enrichment of dynamin-2 beneath the apical surface (Fig. 8, A and B). Dynamin-2 drives vesicle scission at the plasma membrane during endocytosis and is also involved in the secretory pathway, promoting the release of vesicles from the Golgi network and the fusion of vesicles at the apical membrane (Gonzalez-Jamett et al., 2013). Additionally, we stained for protein kinase C and casein kinase substrate in neurons 2 (pacsin-2), an F-BAR protein involved in apical endocytosis (Qualmann and Kelly, 2000); this factor was also increased in tuft cells compared with enterocytes (Fig. 8, C and D).

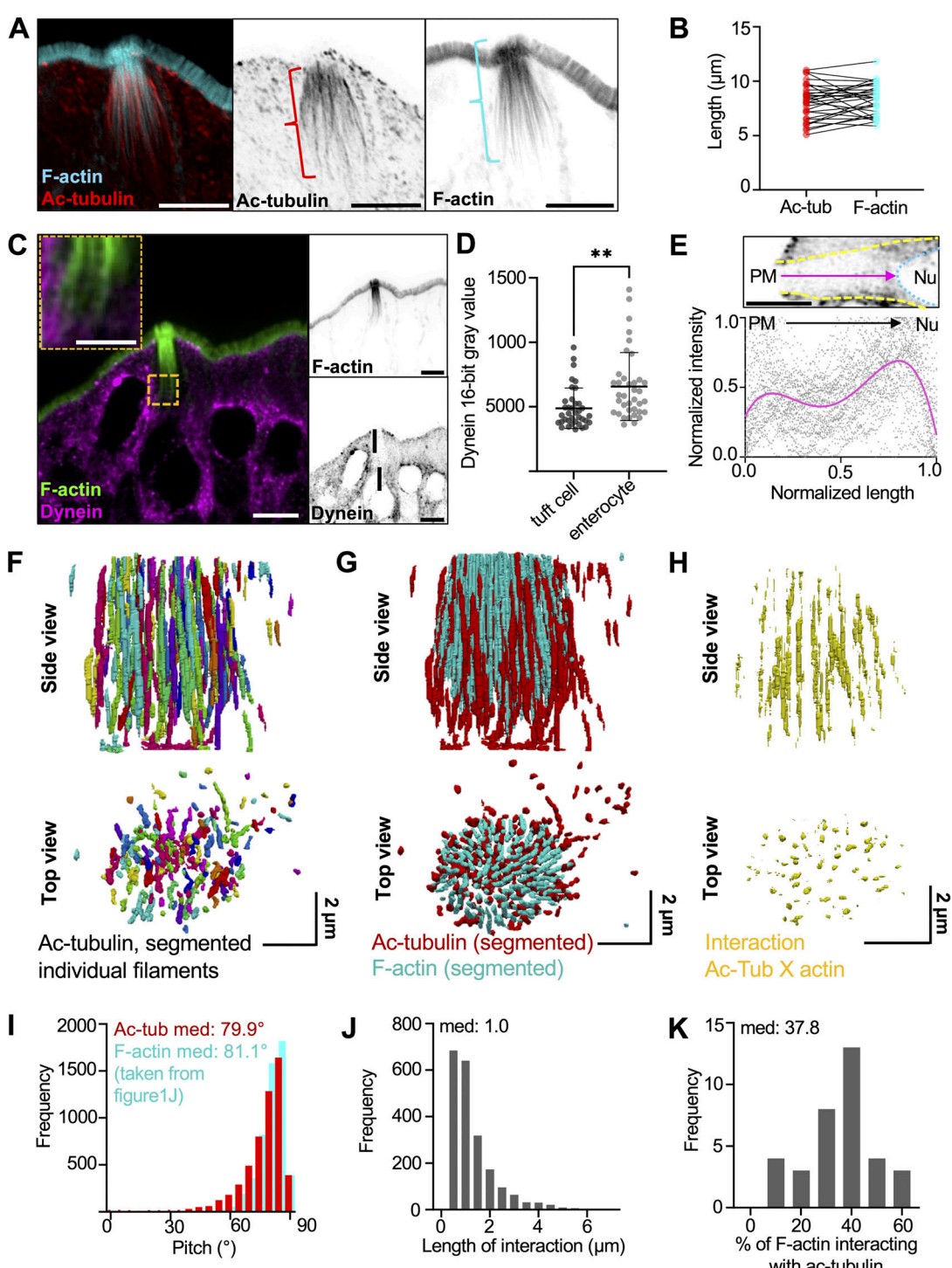

Figure 6. **Core actin bundles co-align and interdigitate with acetylated microtubules. (A)** MaxIP Airyscan image of lateral tissue section (scalebar = 5 μm). **(B)** Graph of length of microtubule network versus actin network in tuft cells, as shown by brackets in A. Paired *t* test, P = 0.3472. **(C)** MaxIP SDC image of a lateral tissue section with dynein immunostaining. Actin marked with phalloidin (scalebar = 5 μm, zoom scalebar = 2 μm). **(D)** Mean dynein intensity measurements taken from sum intensity projections at the apical surface of tuft cells versus neighboring enterocytes, unpaired two-sided *t* test, P = 0.0013, error bars denote mean ± SD (*n* = 37 tuft cells over 3 mice). **(E)** Graph of linescan showing dynein intensity taken from MaxIP images at the apical surface of tuft cell to the top of nucleus. Image above depicts route of linescan (PM = plasma membrane, Nu. = nucleus, scalebar = 5 μm). Raw data shown in gray, and line fit (fourth-order polynomial) in magenta (*n* = 38 tuft cells over 3 mice). **(F)** 3D projection of acetylated tubulin network in a tuft cell using Trainable WEKA segmentation. **(G)** 3D projection of both actin and acetylated microtubule networks, taken from trainable WEKA segmented data. **(H)** Interaction between actin and microtubules considered as areas of overlap or immediate adjacency between both cytoskeletal networks and created via dilation of core actin bundles (×4) using the segmented data in G. **(I)** Frequency diagram showing pitch of microtubules using segmentation from F, data on actin pitch taken from Fig. 1 J (*n* = 35 tuft cells over 3 mice). **(J)** Frequency diagram of the length of individual interactions from H (*n* = 35 tuft cells over 3 mice). **(K)** Frequency diagram of the percentage of total actin within a tuft cell that is interacting with microtubules. Calculated as the total length of interaction (J) divided by the total length of actin per tuft cell (not shown).

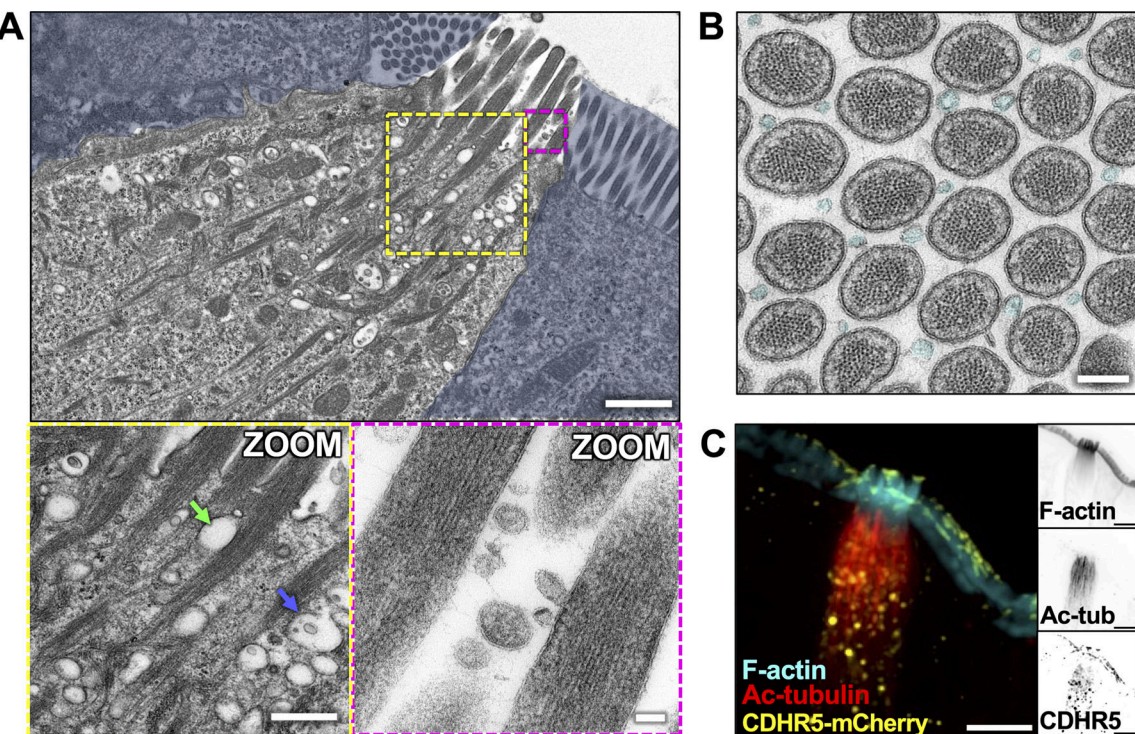

**Figure 7.** **Membranous organelles are associated with the tuft cell cytoskeletal network. (A)** TEM of ultrathin tissue slice showing lateral tuft cell section (scalebar = 1 μm) with enterocytes masked in blue. Zoom inset (left) shows vesicles along core bundles near the apical surface (scalebar = 400 nm). Blue arrow points to an MVB, green arrow points to an electron-lucent vesicle. Zoom inset (right) shows EVs between apical protrusions (scalebar = 50 nm). **(B)** TEM of ultrathin tissue slice showing en face section of the apical tuft showing EVs (cyan) between protrusions (scalebar = 200 nm). This image was taken of the same tuft cell shown in Fig. 2 A. **(C)** MaxIP SDC image of tuft cell from CDHR5-mCherry mouse, immunostained for mCherry to boost the signal and acetylated tubulin (scalebar = 5 μm).

The increased endocytic secretory pathway markers in tuft cells and the close physical contacts between vesicles and core bundles throughout the sub-apical region led us to wonder whether tuft cells express unconventional myosin motors that could power vesicle transport along these structures. Indeed, staining revealed that myosin-6, the sole pointed-end directed myosin with roles in endocytosis (Buss et al., 2001), was enriched in tuft cells compared with enterocytes (Fig. 8, E and F). Myosin-5b, a plus-end directed motor linked to the recycling endosome (Lapierre et al., 2001), was present in both tuft cells and enterocytes at similar levels and demonstrated strong sub-apical accumulation (Fig. 8, G and H). Interestingly, myosin-7b, which associates with the intermicrovillar adhesion complex (Weck et al., 2016), accumulated at the distal tips of tuft protrusions at higher levels than neighboring enterocyte microvilli (Fig. 8, I and J). Together, these data suggest that tuft cells exhibit elevated trafficking activities and further imply that the tuft plays a fundamental role in transferring materials to and/or from the extracellular space.

## Discussion

Whereas the physiological roles of apical specializations such as the brush border and hair bundle are well established (Danielsen and Hansen, 2008; Park and Bird, 2023), how the apical tuft supports the functions of the tuft cell remains unclear. We

reasoned that by developing insight into the architecture of the tuft and its underlying cytoskeleton, we could begin to hypothesize how this structure might be leveraged in vivo to promote tuft cell function and intestinal homeostasis. Previous studies on tuft cells in diverse tissues stopped short of providing the specific molecular and structural details needed to understand the cell biological function of the tuft (Hofer and Drenckhahn, 1992; Sato and Miyoshi, 1997; Trier et al., 1987). To fill these gaps, we combined light and electron microscopy to build detailed maps of the cytoskeletal structures and associated binding proteins that comprise the tuft. Below, we discuss observations on the organization and composition of these structures and their implications for understanding the subcellular function of the tuft.

**Organization of protrusions and core actin bundles in the tuft**
Tuft protrusions extend from the apical surface in a cluster with a median packing angle of ~59° when viewed en face, indicative of tight, hexagonal packing. For cylindrical objects, a hexagonal arrangement allows the cell to achieve maximum packing density. However, the spread of packing angles observed throughout the tuft was large (range of 20°–120°) revealing flexibility in how core actin bundles are situated relative to their neighbors. How actin bundles are positioned and organized during tuft cell differentiation remains unclear. Enterocyte microvilli exhibit near-perfect hexagonal packing, which is enforced by

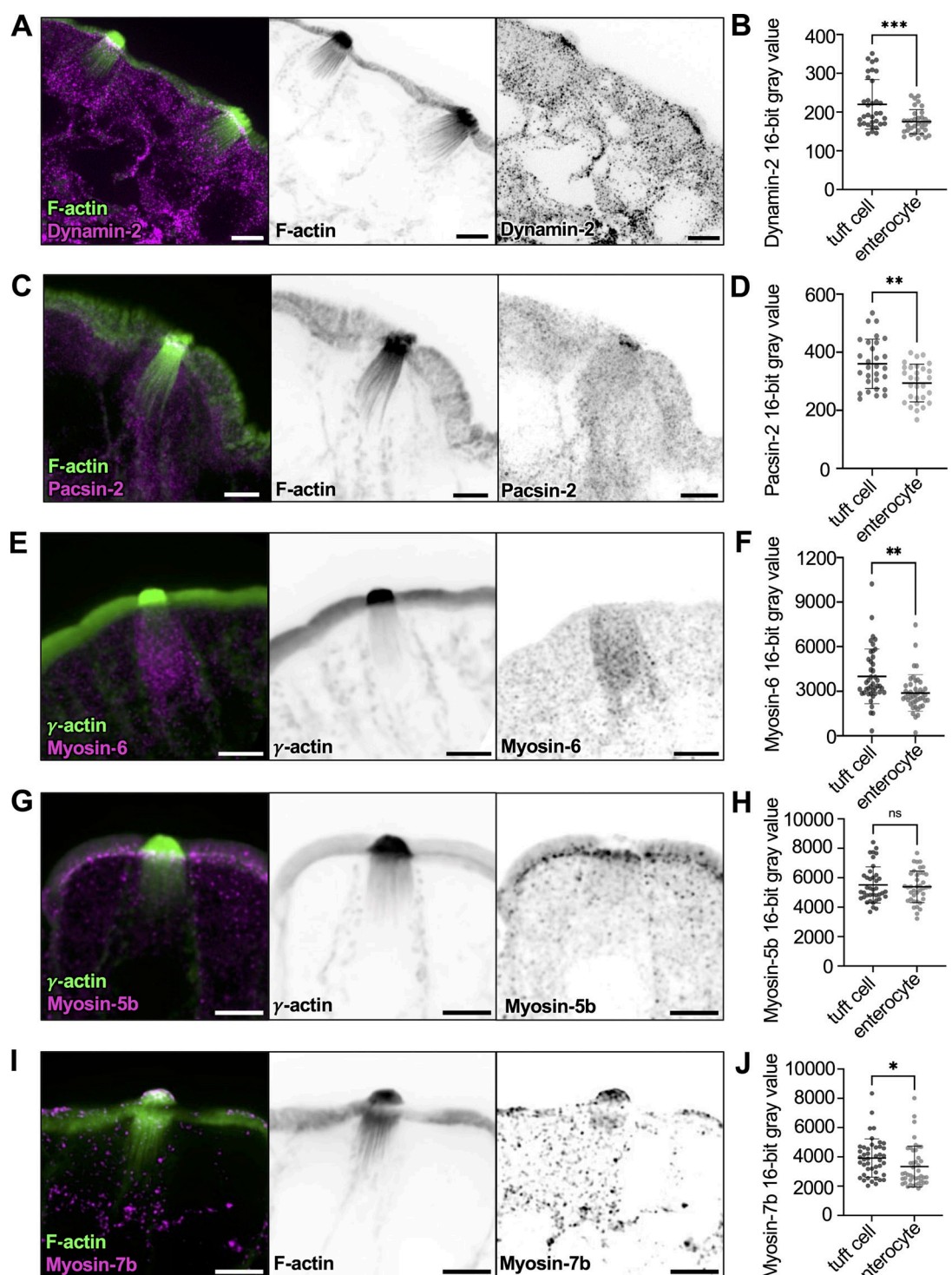

Figure 8. **Tuft cells exhibit higher levels of trafficking machinery relative to enterocytes. (A and C)** MaxIP SDC image of lateral frozen tissue section immunostained for dynamin-2 or pacsin-2, respectively. F-actin marked with phalloidin (scalebar = 5 µm). **(B and D)** Mean intensity measurements taken from sum intensity projections of dynamin-2 (unpaired two-sided *t* test, P = <0.001, *n* = 28 tuft cells over 3 mice) and pacsin-2 (unpaired two-sided *t* test, P = 0.0015, *n* = 34 tuft cells over 3 mice) in the apical surface of tuft cells versus enterocytes. Error bars denote mean ± SD. **(E)** MaxIP SDC image of lateral paraffin-embedded tissue section immunostained with myosin-6. Actin marked by γ-actin staining (scalebar = 5 µm). **(F)** Mean intensity measurements taken from sum intensity projections of myosin-6 (unpaired two-sided *t* test, P = 0.0014) using a line drawn from the apical surface of the cell to top of the nucleus in tuft cells versus enterocytes. Error bars denote mean ± SD. **(G and I)** MaxIP SDC image of lateral tissue section immunostained for (G) Myosin-7b (frozen tissue section, with F-actin marked by phalloidin) and (I) Myosin-5b (paraffin-embedded tissue with actin marked by γ-actin staining). Scalebar = 5 µm. **(H and J)** Mean intensity measurements are taken from sum-intensity projections of Myosin-5b (unpaired two-sided *t* test, P = 0.621, *n* = 39 tuft cells over 3 mice) and Myosin-7b (unpaired two-sided *t* test, P = 0.045, *n* = 44 tuft cells over 3 mice) in the apical surface of tuft cells versus enterocytes. Error bars denote mean ± SD.

protocadherin-based intermicrovillar adhesion complexes (IM-ACs) that localize to the distal tips of these protrusions during and after differentiation (Crawley et al., 2014). Interestingly, some IMAC components, such as CDHR5 and MYO7B, are expressed in tuft cells and positioned to serve a similar role (Fig. 7 C and Fig. 8 I). Whether tuft cells express and localize a full complement of IMAC proteins to control the packing of tuft cell protrusions remains an open question for future studies.

## Tuft core actin bundle dimensions

Core actin bundles in the tuft extend up to ~10 μm, from the distal tips of supported microvillus-like protrusions, down to the rootlet ends that terminate deep in the cytoplasm. Tuft core bundles are longer and contain approximately fivefold more filaments relative to actin cores that support enterocyte microvilli (Mooseker and Tilney, 1975). However, these structures are shorter than some of the more extreme cases of actin bundle assembly. Indeed, *Limulus* sperm extend a long well-ordered actin bundle over 55 μm as part of the acrosomal reaction during fertilization (Tilney, 1975). Moreover, certain forms of stereocilia found on the apex of mechanosensory epithelial cells range up to 120 μm (Manor and Kachar, 2008). Finally, the algae *Nitella* and *Chara* contain internodal cells that position chloroplasts along arrays of polarized actin bundles, extending hundreds of micrometers (Kersey and Wessells, 1976). Future studies might leverage molecular commonalities between these structures to deduce mechanisms that are important for the growth of large-scale actin-based features.

## Filament organization in tuft core actin bundles

Previous studies highlighted EPS8 as a barbed-end binding protein that is enriched at the distal tips of actin bundle supported protrusions including microvilli, stereocilia, and filopodia (Disanza et al., 2006; Gaeta et al., 2021; Manor et al., 2011). Although we detected low levels of EPS8 at the distal tips of tuft protrusions, a structurally related family member, EPS8L2, strongly localized to the tips (Fig. 3), suggesting that filaments within core bundles are uniformly oriented (i.e., polarized) in a barbed-end out configuration. The strong distal end enrichment of EPS8L2 is also consistent with a model where individual filaments extend continuously through the full length of the core bundle, although TEM confirmation of this point was complicated by the tilt of these structures throughout the tuft, which made it difficult to capture their full length in ultrathin sections.

TEM imaging of core bundle cross-sections revealed that these structures are composed of ~100 hexagonally packed actin filaments. This highly ordered pattern contrasts with the F-actin core found in hair cell stereocilia, which exhibit "fluid" packing, characterized by tight but irregular spacing of filaments (Krey et al., 2016). The hexagonal packing observed in tuft cell core bundles might reflect the kinetics or mechanism of bundle polymerization as highly ordered structures are generally produced through slow growth and low actin bundler concentrations (Stokes and DeRosier, 1991). The details of filament packing are important as they may ultimately dictate the mechanical flexibility of the bundle in response to lateral forces (DeRosier et al., 1980). In stereocilia, filaments are crosslinked with multiple structurally distinct actin bundlers, and fluid packing is the result of filaments accommodating these different crosslinker lengths (Krey et al., 2016). Indeed, the knockout of the most abundant crosslinker in hair cells, plastin-1, shifted filament organization from fluid to hexagonal packing (Krey et al., 2016). Based on these results, the ordered packing of filaments in tuft core actin bundles could be dominated by the activity of a single actin crosslinker or instead, multiple distinct crosslinkers of the same length.

## Regionalization of actin-binding proteins along the core bundle axis of the tuft

Our immunofluorescence imaging studies detected several actin bundlers in tuft core bundles, including advillin, pGirdin, fimbrin, espin, LIMA1, and α-actinin 4 (Figs. 4 and S4). Our analysis does not allow us to rank-order the abundance of the factors we examined. However, these proteins did exhibit an unexpected regionalization along the core bundle axis, with pGirdin and espin occupying the membrane-wrapped distal ends; advillin, fimbrin, and α-actinin 4 present in the apical protrusions with decreased signal along the rootlets; and LIMA1 demonstrating restricted localization to the more proximal, rootlet ends. Thus, filament packing patterns could be dictated locally by the specific complement of bundlers present in that segment. Such regionalization would also allow the tuft cell to create segment-specific mechanical properties, which may be important for function. The rootlet localization of LIMA1 is reminiscent of MISP, an actin-bundling protein that targets specifically the rootlets of enterocyte microvilli (Morales et al., 2022). Loss of MISP function in an epithelial culture model shortened rootlets, whereas MISP overexpression led to rootlet hyper-elongation through a mechanism involving competition with Ezrin (Morales et al., 2022). Interestingly, previous in vitro studies showed that LIMA1 holds actin cross-linking activity, slows F-actin depolymerization, and inhibits binding of the Arp2/3 complex (Maul et al., 2003). Mechanistic follow-up studies will be needed to define the function of LIMA1 in shaping tuft core bundles.

## A cytoskeletal superstructure supports the tuft

Our 3D reconstructions revealed that microtubules and core actin bundles interdigitate and exhibit striking co-alignment throughout the sub-apical cytoplasm. Such co-alignment is highly suggestive of specific molecular cross-linking between the two cytoskeletal networks, which could constrain bundle spreading in the sub-apical compartment (Fig. 1, G and H). Physical contact between microtubules and actin networks is known to be critical for normal cell function in a range of tissue contexts (Rodriguez et al., 2003). Neuronal growth cones offer a striking example; in this system, crosslinking between the two cytoskeletons is essential for neurite outgrowth, growth cone motility, and steering (Schaefer et al., 2008). However, in that case, a central bundle of microtubules connects with a peripheral meshwork of F-actin that is highly branched. Neuronal axons, in contrast, contain regions of actin and microtubule co-alignment (Kevenaar and Hoogenraad, 2015), although on a much smaller spatial scale relative to the tuft superstructure. As

previous studies also identified intermediate filaments in the sub-apical domain of tuft cells in other epithelial tissues (Kasper et al., 1994; Luciano et al., 2003), future studies will need to identify the full complement of intermediate filaments expressed in intestinal tuft cells, as well as their organization and positioning relative to the superstructure.

### Potential functions for the cytoskeletal superstructure

The core actin bundles that comprise the tuft extend their rootlets many microns through the sub-apical cytoplasm. Such cytoskeletal architecture is rare in animal cells, although it does point to exciting possibilities for potential functions. For example, in any cytoskeletal network, the net orientation of polymers dictates the direction of motor protein-driven transport (Sivaramakrishnan and Spudich, 2009); polarized (i.e., uniform) filament orientation allows for unidirectional movement of motors, whereas mixed filament polarity can lead to motor stalling as demonstrated in vitro for MYO10 (Nagy et al., 2008). From this perspective, the parallel and polarized barbed-end out orientation of actin filaments in tuft core bundles seem well suited for supporting efficient myosin-driven transport. MyTH4-FERM domain containing myosins are believed to drive transport within the confines of surface protrusions such as filopodia (MYO10) (Berg and Cheney, 2002), microvilli (MYO7B) (Belyantseva et al., 2005; Weck et al., 2016), and stereocilia (MYO7A, MYO15A) (Belyantseva et al., 2005; Weck et al., 2016). Our staining analysis did reveal that MYO7B is highly expressed in tuft cells and accumulates at the distal tips of tuft core bundles; such localization is typically viewed as a telltale sign of barbed-end directed motor activity (Berg and Cheney, 2002). MYO5B and MYO6 are also expressed in tuft cells and are well-positioned to use the superstructure as a track for cargo transport. This idea is consistent with previous studies, which implicate these myosins in vesicle trafficking events (e.g., recycling and endocytosis) that are expected to take place in the sub-apical compartment occupied by the superstructure (de Jonge et al., 2019; Engevik and Goldenring, 2018).

The cytoskeletal superstructure might also support directed transport by microtubule motors, which have well-established roles in cytoplasmic trafficking (Hirokawa et al., 2009). In this scenario, the long rootlets of core actin bundles might simply serve as a physical scaffold for the alignment of stable microtubules. As some myosins directly interact with kinesins and such interactions increase the movement of both motors along networks containing F-actin and microtubules (Ali et al., 2008), cooperation between these systems might also be possible in the cytoskeletal superstructure that supports the tuft.

What cargoes might be transported along the tuft cytoskeleton? Tuft cells are responsible for the secretion of several effectors including IL-25, CysLTs, PGD2, and Ach, (Labed et al., 2018; McGinty et al., 2020; Oyesola et al., 2021; von Moltke et al., 2016) which could be potential cargoes. EM and light microscopy analysis revealed large numbers of vesicles with diverse morphologies distributed along core bundle rootlets. In some cases, these were identifiable as MVBs, leading to the possibility that these organelles are trafficked apically, along rootlets, to eventually release their vesicle cargoes from the cell surface. The abundance of EVs observed between individual tuft protrusions is consistent with this idea. Live imaging of tuft cells expressing labeled versions of secreted effectors or MVB markers will be needed to determine if the cytoskeletal superstructure in the tuft supports these trafficking functions.

### Next steps

Although previous studies uncovered general characteristics of tuft cells, the quantitative morphometry we report here provides a framework for future mechanistic studies of tuft cell function. Additionally, our work provides an experimental blueprint and quantitative point of comparison for the eventual characterization of tuft cells in other tissues, as well as neuron-like and immune-like tuft cell subtypes. Tuft cells have been increasingly implicated in intestinal health for parasite clearance as well as restoration of intestinal barrier function in models of Crohn's disease and ulcerative colitis (Banerjee et al., 2020). Thus, investigation of how the unique tuft cell cytoskeletal features identified here are perturbed in these disease states should be an important goal for future studies.

## Materials and methods

All authors had access to the study data and reviewed and approved the final manuscript.

### Experimental model details

#### Cell culture model

HeLa cells were cultured at 37°C and 5% $CO_2$ in Dulbecco's modified Eagle's medium (#10-013-CV; Corning) with high glucose and 2 mM L-Glutamine supplemented with 10% fetal bovine serum. Transfections were performed using Lipofectamine 2000 (#11668019; Thermo Fischer Scientific) according to the manufacturer's protocol. Cells were replated onto 35-mm glass-bottom dishes or coverslips (#D35-20-1.5-23 N; Cellvis) and incubated in Lipofectamine overnight. Media was replaced the following morning and cells were washed gently in PBS.

#### Mouse models

Animal experiments were carried out in accordance with Vanderbilt University Medical Center Institutional Animal Care and Use Committee (IACUC) guidelines under IACUC Protocols. NM2C-EGFP mice were obtained as a generous gift from Dr. Robert Adelstein (National Heart, Lung, and Blood Institute, National Institutes of Health [NIH]) and were described previously (Ebrahim et al., 2013). Mice expressing endogenously tagged CDHR5 mice were created in collaboration with the Vanderbilt Genome Editing Resource. Briefly, CRISPR/Cas9 genome engineering methods were used to insert a flexible linker (5′AGCGGCGGAGGTAGCGGAGGTGGCAGC-3′) and mCherry coding sequence at the 3′ end of the CDHR5 terminal coding exon.

### Method details

#### Succinate administration

For tissue used in TEM preparation, 120 nM of sodium succinate hexahydrate (with dextrose at 1%/vol to improve taste) was

added to drinking water of wild-type (C57BL6) mice for 1 wk prior to euthanasia and tissue preparation.

### Frozen and whole-mount tissue preparation

Segments of the proximal intestine were removed, flushed with PBS, and prefixed for 15 min with 4% paraformaldehyde (PFA) to preserve tissue structure. The tube was then cut along its length, subdissected into 0.5-mm chunks, fixed for an additional 30 min in 4% PFA at room temperature, and washed three times in PBS. Whole-mount samples were then moved to Eppendorf tubes for staining. For frozen sections, tissue samples were gently placed on top of a 30% sucrose solution in TBS and allowed to sink to the bottom overnight at 4°C. Specimens were then swirled in three separate blocks of OCT (Electron Microscopy Sciences), oriented in a block filled with fresh OCT, and snap-frozen in dry ice-cooled acetone. Samples were cut into 10-µm sections and mounted on slides for staining.

### Immunofluorescence

***Cell culture.*** For spinning disk confocal imaging, cells were washed three times with prewarmed PBS before being fixed in 4% PFA (#15710; Electron Microscopy Sciences) in PBS for 15 min at 37°C. Cells were then incubated for 1 h with Alexa Fluor 568 phalloidin (1:200; #A12380; Invitrogen) at room temperature. Coverslips were then washed three times in PBS and mounted on slides with 20 µl ProLong Gold (#P36930; Invitrogen).

***Frozen tissue sections.*** For Zeiss Airyscan or spinning disk confocal imaging, frozen tissue sections were rinsed three times in PBS and permeabilized with 0.1% Triton X-100 in PBS for 15 min at room temperature. After permeabilization, tissue slides were rinsed three times in PBS and blocked with 10% BSA (#9048-46-8; Research Products International) in PBS for 2 h at 37°C in a humidified chamber. Immunostaining was performed using primary antibodies (see Table 1 for details) diluted in 1% BSA overnight at 4°C. After incubation with primary antibody, slides were rinsed three times in PBS and incubated for 2 h with appropriate secondary antibodies (see Table 1 for details) and an Alexa Flour conjugated phalloidin or γ-actin (see Table 1 for details) at room temperature. Slides were then washed three times with PBS and coverslips were mounted with 60 µl Pro-Long Gold (#P36930; Invitrogen) overnight.

***Whole-mount tissue sections.*** For spinning disc confocal imaging, whole-mount tissue was stained in Eppendorf tubes. The tissue was rinsed three times in PBS and permeabilized with 0.2% Triton X-100 in PBS for 30 min at room temperature on a rocker. After permeabilization, tissue was rinsed three times in PBS and blocked in 5% BSA overnight at 4°C on a rocker. Primary antibody (see Table 1 for details) diluted in 1% BSA was added and left overnight at 4°C on a rocker. The next day, the tissue was washed three times in PBS and the appropriate secondary antibodies (see Table 1 for details) were diluted in 1% BSA for 4 h at room temperature on a rocker. After 2 h of incubation, an Alexa Fluor conjugated phalloidin was added to the tissue (see Table 1 for details). The tissue was then washed three times in PBS and mounted villi-down on a glass-bottom dish with 20 µl ProLong Gold with a coverslip on top overnight.

Table 1. **Reagents and resources**

| Reagent or resource | Source | Identifier |
|---|---|---|
| Antibodies | | |
| Anti-EPS8 (mouse) (1:400) | Sigma-Aldrich | Cat# HPA003897; RRID: AB_1848224 |
| Anti-Eps8L2 (rabbit) (1:400) | Sigma-Aldrich | Cat# HPA041143; RRID: AB_10794985 |
| Anti-ZO-1 (rat) (1:100) | Milliporesigma | Cat# MABT11; RRID: AB_10616098 |
| Anti-NM2A (rabbit) (1:500) | BioLegend | Cat# 909802; RRID: AB_2734686 |
| Anti-γ-actin conjugated AF 647 (mouse) (1:100) | Santa Cruz Biotechnology | Cat# sc-65638 AF647; RRID: AB_2890621 |
| Anti-CAMSAP2 (rabbit) (1:200) | Novus | Cat# NBP1-21402; RRID: AB_1659977 |
| Anti-CAMSAP3 (rabbit) (1:100) | Thermo Fisher Scientific | Cat# PA5-48993; RRID: AB_2634449 |
| Anti-KIF1C (rabbit) | Cytoskeleton | Cat# AKIN11, RRID: AB_10707922 |
| Anti-myosin-5b (rabbit) (1:200) | Novus | Cat# NBP1-87746; RRID: AB_11034537 |
| Anti-myosin-7b (rabbit) (1:50) | Sigma-Aldrich | Cat# HPA039131; RRID: AB_10672771 |
| Anti-myosin-6 (rabbit) (1:100) | Thermo Fisher Scientific | Cat# PA5-57290, RRID: AB_2644375 |
| Anti-Girdin (Y1798) (rabbit) (1:100) | IBL—America | Cat# 28143, RRID: AB_3249611 |
| Anti-α-actinin 4 conjugated AF 647 (rabbit) (1:100) | Abcam | Abcam Cat# ab198610; RRID: AB_2890654 |
| Anti-fimbrin (mouse) (1:50) | Santa Cruz Biotechnology | Cat# sc-271223, RRID: AB_10614853 |
| Anti-Espin (rabbit) (1:500) | Thermo Fisher Scientific | Cat# PA5-5594; RRID: AB_2641126 |
| Anti-Advillin (rabbit) (1:500) | Thermo Fisher Scientific | Cat# PA5-63621; RRID: AB_2638403 |
| Anti-LIMA1 (rabbit) (1:500) | Sigma-Aldrich | Cat# HPA052645; RRID: AB_2681898 |
| Anti-myosin-1b (rabbit) (1:500) | Thermo Fisher Scientific | Cat# PA5-63820; RRID: AB_2644364 |
| Anti-Ezrin (rabbit) (1:200) | Cell Signaling | Cat# 3145S; RRID: AB_2100309 |
| Anti-pEzrin (T567) (rabbit) | Cell Signaling | Cat# 3726S; RRID: AB_10560513 |
| Anti-Ac-tubulin (mouse) (1:800) | Sigma-Aldrich | Cat# T6793; RRID: AB_477585 |
| Anti-dynein (mouse) (1:50) | Thermo Fisher Scientific | Cat# MA1-070; RRID: AB_2093668 |
| Anti-mCherry (rat) (1:100) | Thermo Fisher Scientific | Cat# M11217; RRID: AB_2536611 |
| Anti-pacsin-2 (1:200) | Sigma-Aldrich | Cat# HPA049854; RRID: AB_2680915 |
| Anti-dynamin-2 (1:200) | Sigma-Aldrich | Cat# HPA054246; RRID: AB_2682427 |

| Reagent or resource | Source | Identifier |
|---|---|---|
| Goat anti-mouse Alexa Fluor 488 F(ab′) 2 fragment (1:1,000) | Thermo Fisher Scientific | Cat# A-11017; RRID: AB_2534084 |
| Goat anti-rabbit Alexa Fluor 488 F(ab′) 2 fragment (1:1,000) | Thermo Fisher Scientific | Cat# A-11070; RRID: AB_2534114 |
| Goat anti-rabbit Alexa Fluor 568 F(ab′) 2 fragment (1:1,000) | Thermo Fisher Scientific | Cat# A-21069; RRID: AB_2535730 |
| Chemicals, peptides, and recombinant proteins | | |
| Alexa Fluor Plus 405 Phalloidin (1:200) | Thermo Fisher Scientific | Cat# A30104 |
| Alexa Fluor 647 Phalloidin (1:200) | Thermo Fisher Scientific | Cat# A22287 |
| Alexa Fluor 568 Phalloidin (1:200) | Thermo Fisher Scientific | Cat# A12380 |
| 16% PFA | Electron Microscopy Sciences | Cat# 15710 |
| Triton X-100 | Sigma-Aldrich | Cat# T8787 |
| ProLong Gold Antifade Reagent | Invitrogen | Cat# P36930 |
| Lipofectamine 2000 | Thermo Fisher Scientific | Cat# 11668019 |
| Antibiotic-antimycotic (anti-anti) | Gibco | Cat# 15240062 |
| Glutaraldehyde 25% | Electron Microscopy Sciences | Cat# 16220 |
| Tannic acid | Electron Microscopy Sciences | Cat# 21700 |
| Osmium tetroxide | Electron Microscopy Sciences | Cat# 19112 |
| Cell lines | | |
| HeLa | American Type Culture Collection | Cat# CCL2 |
| Mouse models | | |
| C57BL6 | | |
| CDHR5-mCherry mice | Developed in-house by the Vanderbilt Genome Editing Resource | N/A |
| NM2C-EGFP mice | Generous gift from the Adelstein laboratory (NHLBI, NIH) as described in Ebrahim et al. (2013) | N/A |
| Recombinant DNA | | |
| EPS8L2-EGFP | Tyska laboratory | N/A |
| Software and algorithms | | |
| FIJI | https://fiji.sc/ | N/A |
| NIS AR Elements analysis | Nikon (https://www.microscope.healthcare.nikon.com/products/software/nis-elements/nis-elements-advanced-research) | N/A |

| Reagent or resource | Source | Identifier |
|---|---|---|
| Prism 9 | GraphPad (https://graphpad.com) | N/A |
| MATLAB | https://www.mathworks.com/products/matlab.html | N/A |

### Light microscopy

Laser scanning confocal microscopy was performed on a Nikon A1 microscope equipped with 488, 568, and 647 nm lasers and a 100×/1.49 NA TIRF oil immersion objective. Spinning disk confocal imaging was conducted using a Nikon Ti2 inverted light microscope with a Yokogawa CSU-W1 spinning disk head, a Photometrics Prime95B sCMOS camera, four excitation lasers (488, 568, 647, and 405 nm), and a 100×/1.49 NA TIRF oil immersion objective or a 60×/1.49 NA TIRF oil immersion objective. Images presented in the figures were deconvolved (Richardson-Lucy deconvolution of image volumes, 20 iterations) using Nikon Elements software. Super-resolution images were collected on a Zeiss LSM980 Airyscan microscope with four excitation lasers (488, 568, 647, and 405 nm) and a 63×/1.43 NA oil immersion objective; images were processed using Zeiss Zen software.

### EM

To prepare samples for TEM, 1-mm pieces of small intestine were fixed for 1 h in 2% paraformaldehyde, 2% glutaraldehyde, and 0.1 M cacodylate buffer with 2 mM $CaCl_2$. The samples were washed in 0.1 M HEPES and slowly equilibrated with 30% glycerol as a cryoprotectant. Samples were plunge frozen in liquid ethane followed by freeze-substitution in 1.5% uranyl acetate in methanol for 48 h at −80°C. Samples were washed extensively in methanol and infiltrated with HM20 Lowicryl. The HM20 was polymerized by UV light at −30°C for 24 h under nitrogen vapor. Samples were sectioned on a Leica UC7 ultramicrotome with a nominal thickness of 70 nm on 200 mesh Ni grids and poststained with uranyl acetate and lead citrate. Images were collected with a Tecnai T-12 transmission electron microscope operating at 100 kV using an AMT nanosprint5 CMOS camera.

### Electron tomography

Electron tomography was performed on a JEOL 2100+ operating at 200 keV using an AMT nanosprint CMOS camera and SerialEM for automated acquisition. Samples were unidirectionally tiled 60° to −60° at 1° increments at a target defocus of −2 μm. Tilt series were reconstructed using the IMOD/etomo software suite with patch tracking and backplane projection (Mastronarde, 2005). Tomogram stacks were further processed in FIJI using brightness/contrast and summed Z-projection through substacks to remove noise.

### Analysis of scRNA-seq data

We used a previously published scRNA-seq dataset to probe for tuft cell–enriched candidate genes (Banerjee et al., 2020). For

that study, intestinal crypts were isolated using an EDTA-chelation protocol, followed by cold dissociation into single-cell suspensions, and inDrops scRNA-seq thereafter (Simmons and Lau, 2022). For our analysis, we used data from $n = 6$ mice ran as independent experiments. Filtered count matrices were downloaded from NCBI GEO Accession GSE145830. Data were processed with the same analysis pipeline as the original manuscript using Seurat (Stuart et al., 2019). Data were normalized to total transcript count per cell and then arcsinh-transformed onto a log-like scale. Highly variable genes were selected to use for canonical correlation analysis (CCA) to align the independent experiments. CCA coordinates were then used as input for t-distributed stochastic neighbor embedding (t-SNE) for visualization. t-SNE plots were then generated by feeding the processed data into CellxGene (Megill et al., 2021, *Preprint*).

### Image analysis and statistics

All images were processed using Nikon Elements or FIJI software (https://fiji.sc/).

***Actin bundle length and intensity, and localization of actin-binding proteins.*** Lateral images of frozen sections were analyzed using FIJI. Maximum intensity projections (MaxIP) of core actin bundles were generated and lines were drawn from the apical tuft to the bottom of the rootlets to enable measurements of bundle length and marker intensity along the bundle axis.

***Number, angle, and bundle spreading measurements.*** Images captured en face using whole-mount tissue were used for these quantifications. MaxIPs of image volumes acquired just beneath the apical surface were used to count the number of individual core actin bundles per tuft cell. The core actin bundles were identified by dark spot identification in Nikon Elements and angles were measured from every bundle to two adjacent neighbors. Bundle spreading was measured by calculating bundle area versus cell area (identified by phalloidin staining) using individual slices at the apical surface, 1.5, 3, and 4.5 μm below the apical surface.

***Fourier analysis of TEM images.*** To map filaments of equidistant packing or hexagonal packing back onto an original TEM image, Fourier filtering was performed with DigitalMicrograph and Fiji. Target areas were selected and then filtered using a Fast Fourier Transform (FFT). A circular (for equidistant packing) or hexagonal mask (for hexagonal packing) was placed over the predominant reflections in Fourier space using the masking tools built-in DigitalMicrograph, making the mask as tight as possible to reduce the background Fourier signal. After applying the mask, an inverse FFT was performed to generate the corresponding real-space image. This image was thresholded in Fiji to remove the background signal, pseudocolored, and then merged with the original image.

***Trainable WEKA segmentation of core actin bundles and microtubules.*** Oversampled images of en face whole-mount tissues stained with phalloidin and acetylated tubulin were opened in FIJI and subject to Trainable WEKA segmentation (Arganda-Carreras et al., 2017) to identify core actin bundles and microtubules independently. Actin probability maps were put under the minimum auto local thresholding while microtubule probability maps were put under the default auto local threshold.

These thresholds were chosen to maximize the selection of the cytoskeleton. Both maps were opened in Nikon Elements and thresholded so that individual objects (in this case individual core actin bundles or microtubules) could be visualized. From this data, we obtained the pitch (°) data. To determine the interaction between microtubules and core actin bundles, we dilated the core actin bundle signals and measured the length of overlap with microtubules. We further compared the volume of the overlap with the volume of the core actin bundles to determine the percent of actin polymer overlapping with microtubules.

***Intensity analysis of antibody staining.*** For quantification of NM2C intensities in tuft cells and enterocytes, we used FIJI to analyze en face images from whole-mount tissues. Sum intensity projections encompassing the apical surface were generated, a region of interest (ROI) was drawn to capture the tuft cells or a neighboring enterocyte, and the mean intensities were measured. For other intensity measurements, including NM2A, Ezrin, pEzrin, dynamin-2, pacsin-2, dynein, myosin-5b, and myosin-7b, lateral images of frozen tissue sections were analyzed in FIJI. Sum intensity projections were generated to include the full tuft, an ROI was drawn over apical protrusions, (or just under apical protrusions for myosin-5b), and the mean intensity of the marker was measured using an ROI tool. An identically sized ROI was used to measure the mean intensities from neighboring enterocytes. Myosin-6 intensity analysis was conducted similarly but a line scan (thickness of 10) drawn from the apical surface down to the nucleus was used in place of an ROI.

***Nearest neighbor measurements for filaments within individual core actin bundles.*** FIJI plugin TrackMate was used to mark individual actin filaments within single-core actin bundles. The center of each filament was exported as an x,y coordinate, and a custom MATLAB script (created with generative AI) was used to identify: (i) the number of nearest neighboring filaments within a 12-nm radius of each filament, (ii) the distance between nearest neighbors, and (iii) the angle between a filament and two adjacent nearest neighbors.

For all statistics, data distribution was assumed to be normal, but this was not formally tested.

### Online supplemental material

Fig. S1 shows the apical surface area and circularity of the tuft versus enterocyte as well as the distribution of NM2C in the tuft cell. It also shows the t-SNE analysis of scRNA-seq data of both NM2A and NM2C and the intensity of both proteins in tuft cells versus enterocytes based on tissue staining. Fig. S2 shows additional ultrastructure measurements including diameter of apical protrusions, membrane–actin distance, and number of nearest neighbors for filaments within a bundle. This last measurement includes an example bundle from the MATLAB analysis to color-code nearest neighbor distances. Fig. S3 shows t-SNE analysis of scRNA-seq data generated from mouse intestinal tissue of several genes of protein products including EPS8, EPS8L2, ESPIN, LIMA1, α-actinin, myosin-1b, and Ezrin. Fig. S4 shows antibody staining of α-actinin and advillin in tissue. Fig. S5 shows a TEM of a lateral section of a tuft cell highlighting four

areas of microtubules and actin in proximity. It also includes tissue staining for CAMSAP2, CAMSAP3, and KIF1C in tuft cells. Video 1 shows a tomogram of a lateral section of a tuft cell depicting core bundle continuity from protrusion to apical cytoplasm.

## Data availability

All data are available in the published article and its online supplemental material. The custom MATLAB script to quantify nearest neighbors is available upon request.

## Acknowledgments

The authors thank all members of the Tyska laboratory for their feedback. We thank the James Goldenring, Izumi Kaji, and Irina Kaverina laboratories for supplying antibodies used in this study; in particular, we thank Margret Fye, Andreanna Burman, and Avishkar Sawant. We thank the Vanderbilt Genome Editing Resource supported by the Cancer Center Support Grant P30 CA68485 for their assistance in generating the CDHR5-mCherry knockout mouse. Microscopy was performed in part by the Vanderbilt Cell Imaging Shared Resource. We also acknowledge the Translational Pathology Shared Resource supported by National Cancer Institute/NIH Cancer Center Support Grant P30 CA068485 for their assistance in paraffin-embedded tissue sectioning.

This study was supported by the NIH National Institute of Diabetes and Digestive and Kidney Diseases National Research Service Award F31 DK141157 (J.B. Silverman), as well as NIH grants R01 DK103831 (K.S. Lau), R01 DK095811 (M.J. Tyska), R01 DK125546 (M.J. Tyska), and R01 DK111949 (M.J. Tyska).

Author contributions: J.B. Silverman: Conceptualization, Data curation, Formal analysis, Funding acquisition, Investigation, Methodology, Project administration, Validation, Visualization, Writing—original draft, Writing—review & editing, E.E. Krystofiak: Data curation, Formal analysis, Investigation, Methodology, Visualization, Writing—review & editing, L.R. Caplan: Data curation, Formal analysis, Methodology, Software, Writing—review & editing, K.S. Lau: Data curation, Formal analysis, Resources, Software, Visualization, Writing—original draft, Writing—review & editing, M.J. Tyska: Conceptualization, Funding acquisition, Methodology, Project administration, Supervision, Writing—original draft, Writing—review & editing.

Disclosures: The authors declare no competing interests exist.

Submitted: 13 April 2024

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

# Supplemental material

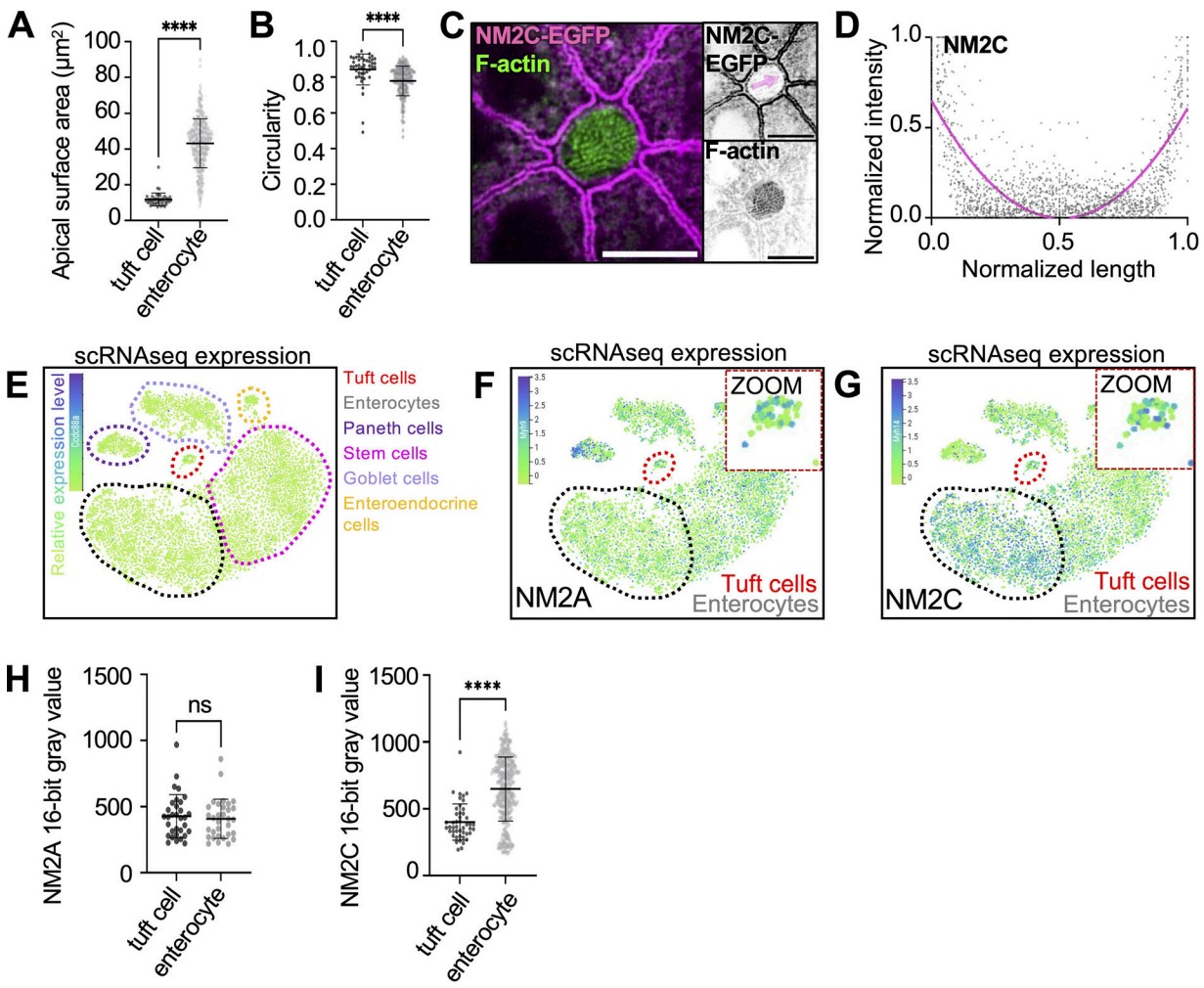

Figure S1. **NM2A and NM2C are present on the lateral apical junctions of tuft cells. (A)** Apical surface area measurement using ZO-1 staining in whole-mount tissue as imaged in Fig. 1 M. Unpaired two-sided *t* test, P < 0.001, error bars denote mean ± SD (*n* = 45 tuft cells over 3 mice). **(B)** Apical circularity measurement using ZO-1 staining in whole-mount tissue as imaged in Fig. 1 M, unpaired two-sided *t* test, P < 0.001, error bars denote mean ± SD (*n* = 45 tuft cells over 3 mice). **(C)** MaxIP SDC image of en face whole-mount NM2C-EGFP tissue, actin stained with phalloidin (scalebar = 5 μm). **(D)** Graph of linescans for NM2C intensity drawn across the apical surface of a tuft cell (magenta arrow in B) in MaxIPs of NM2C-EGFP whole-mount tissue (*n* = 43 tuft cells over 3 mice). **(E–G)** t-SNE analysis of scRNA-seq data generated from mouse intestinal tissue. The common names of the protein products of the genes are labeled on each plot. Heatmap overlay represents the Arcsinh-scaled of normalized transcript count. Tuft cells red circle/zoom and enterocytes gray circle (*n* = 6 mice). **(E)** t-SNE plots showing cell types (dotted outlines). **(F)** t-SNE plot of *myh9* (NM2A). **(G)** t-SNE plot of *myh14* (NM2C). **(H)** Mean NM2A intensity measurements in tuft cells versus enterocytes in frozen tissue sections, unpaired two-sided *t* test, P = 0.6179, error bars denote mean ± SD (*n* = 32 tuft cells over 3 mice). **(I)** Mean NM2C intensity measurement in tuft cells versus enterocytes in whole-mount tissue sections, unpaired two-sided *t* test, P < 0.0001, error bars denote mean ± SD, (*n* = 46 tuft cells over 3 mice).

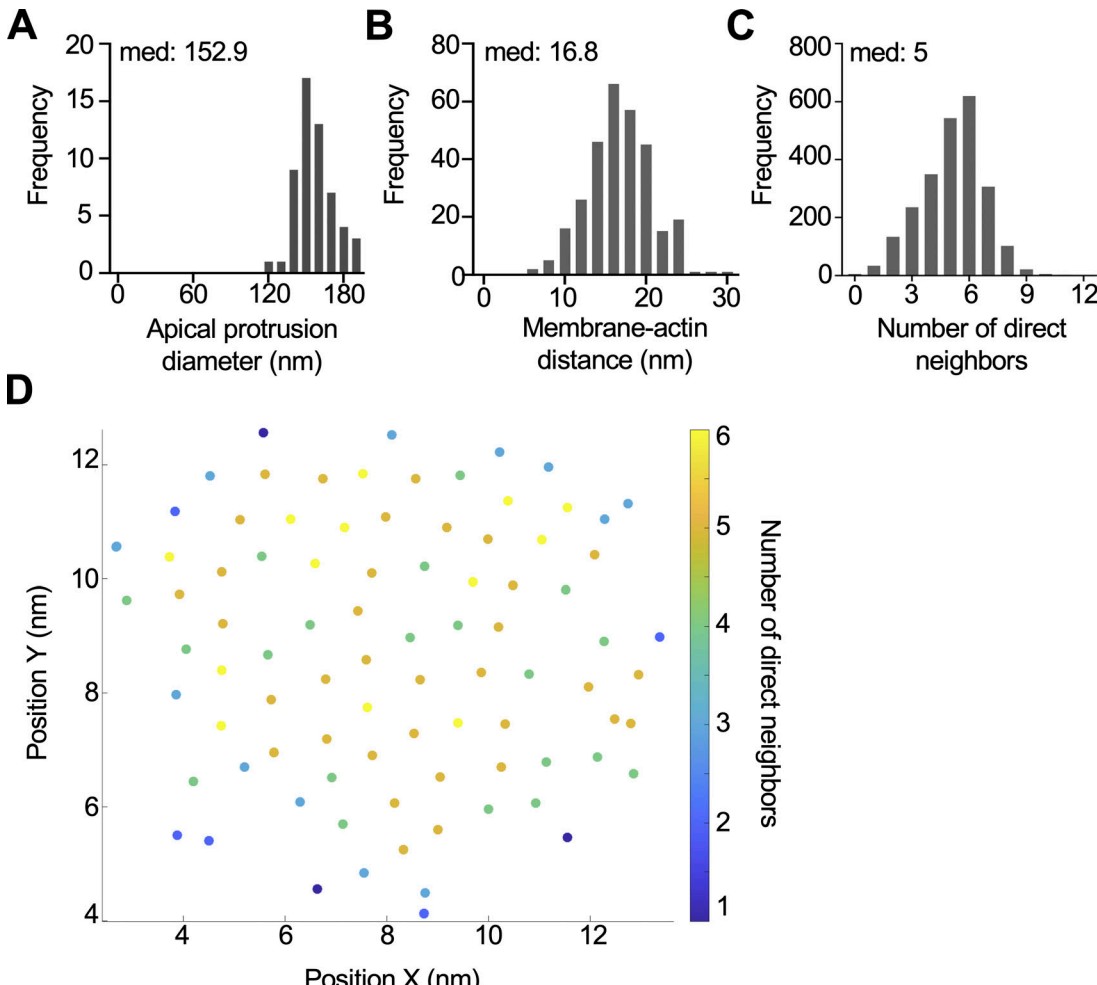

Figure S2. **Core actin bundles contain hexagonally packed filaments. (A)** Frequency diagram of the apical protrusion diameter in tuft cells ($n$ = 57 protrusions over 4 tuft cells). **(B)** Frequency diagram distance between outer protrusion membrane to actin measured at five separate places in each bundle ($n$ = 30 bundles over 3 tuft cells). **(C)** Frequency diagram of the number of nearest filament neighbors (within a 12-nm radius) ($n$ = 22 bundles over 3 tuft cells). **(D)** Example of nearest neighbor analysis showing a single core actin bundle and individual filaments. Filament coordinates were derived using Trackmate and put into a custom MATLAB script to quantify nearest neighbors (within a 12-nm radius). Filaments are color-coded based on the number of nearest neighbors.

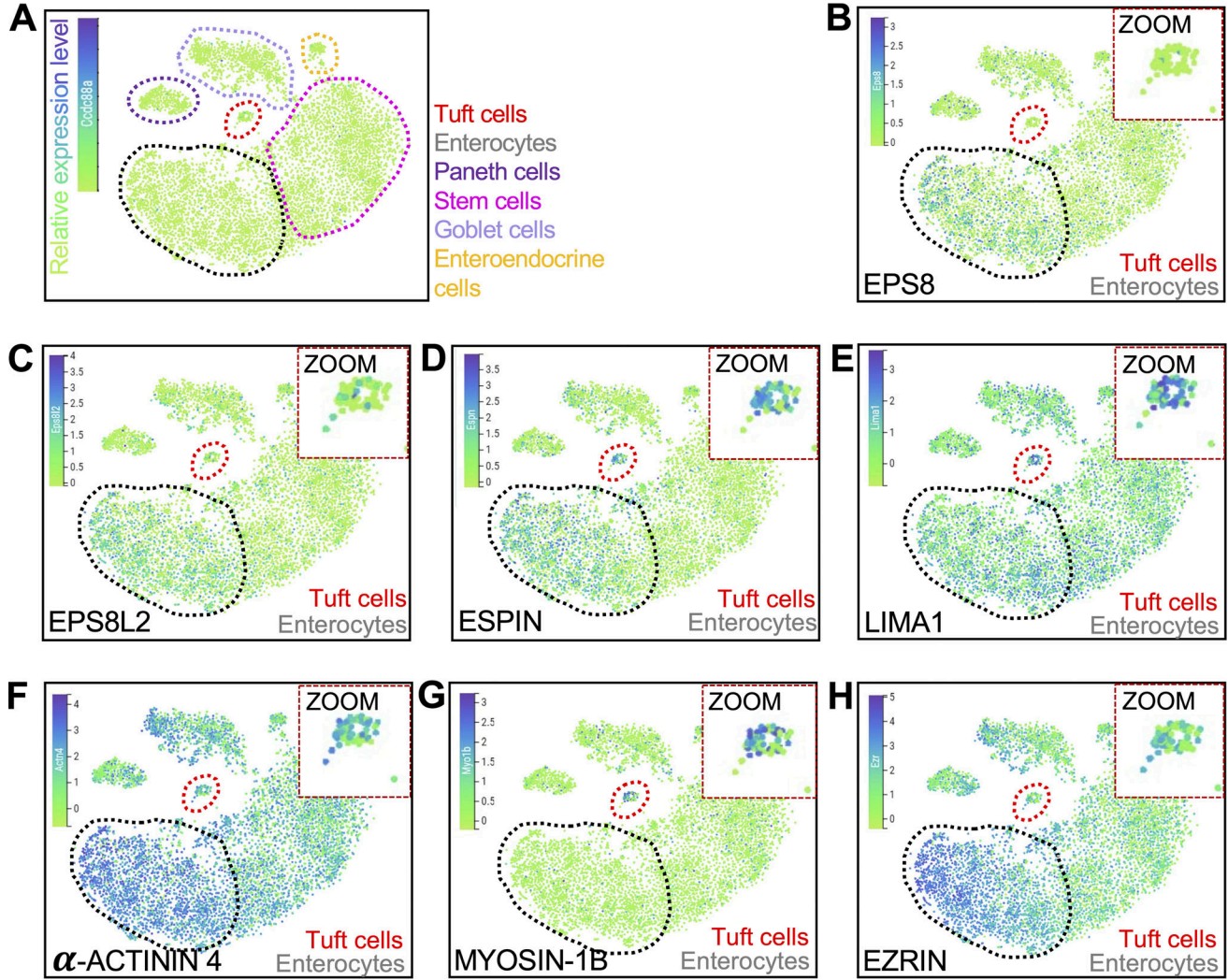

Figure S3.   **t-SNE analysis of scRNA-seq data generated from mouse intestinal tissue. (A–H)** t-SNE plots showing (A) cell types (dotted outlines), (B) *Eps8*, (C) *Eps8l2*, (D) *Espn*, (E) *Lima1*, (F) *Actn4*, (G) *Myo1b*, and (H) *Ezrn*. Measurements in tuft cells—red circle/zoom and in enterocytes—gray circle (*n* = 6 mice). The common names of the protein products of the genes are labeled on each plot. Heatmap overlay represents the Arcsinh-scaled normalized transcript count.

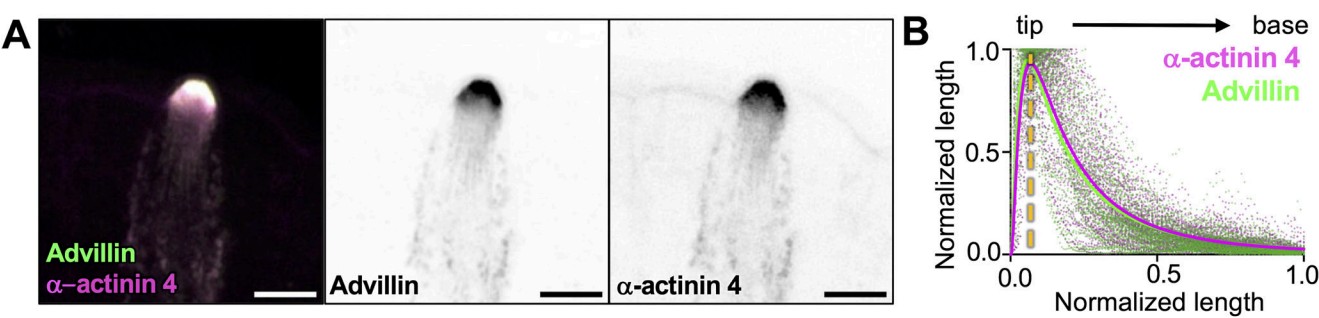

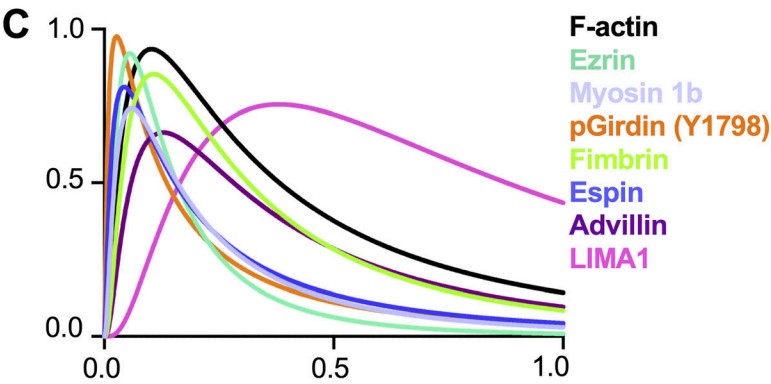

Figure S4. **Distribution of actin-binding proteins along the core bundle axis. (A)** MaxIP SDC image of paraffin-embedded tissue and immunostained for α-actinin 4 and Advillin (scalebar = 5 µm). **(B)** Graph of linescans drawn from MaxIP images from apical tip to the base of core bundle, measuring the intensity of actin-binding proteins. Line fit (lognormal) in magenta for α-actinin 4 and green for advillin with raw values for both proteins in magenta and green, respectively. The yellow line indicates the peak of line fits (*n* = 35 tuft cells over 3 mice). **(C)** Overlayed line fits (lognormal) from actin binding proteins in Fig. 4, B, D, F, H, and J.

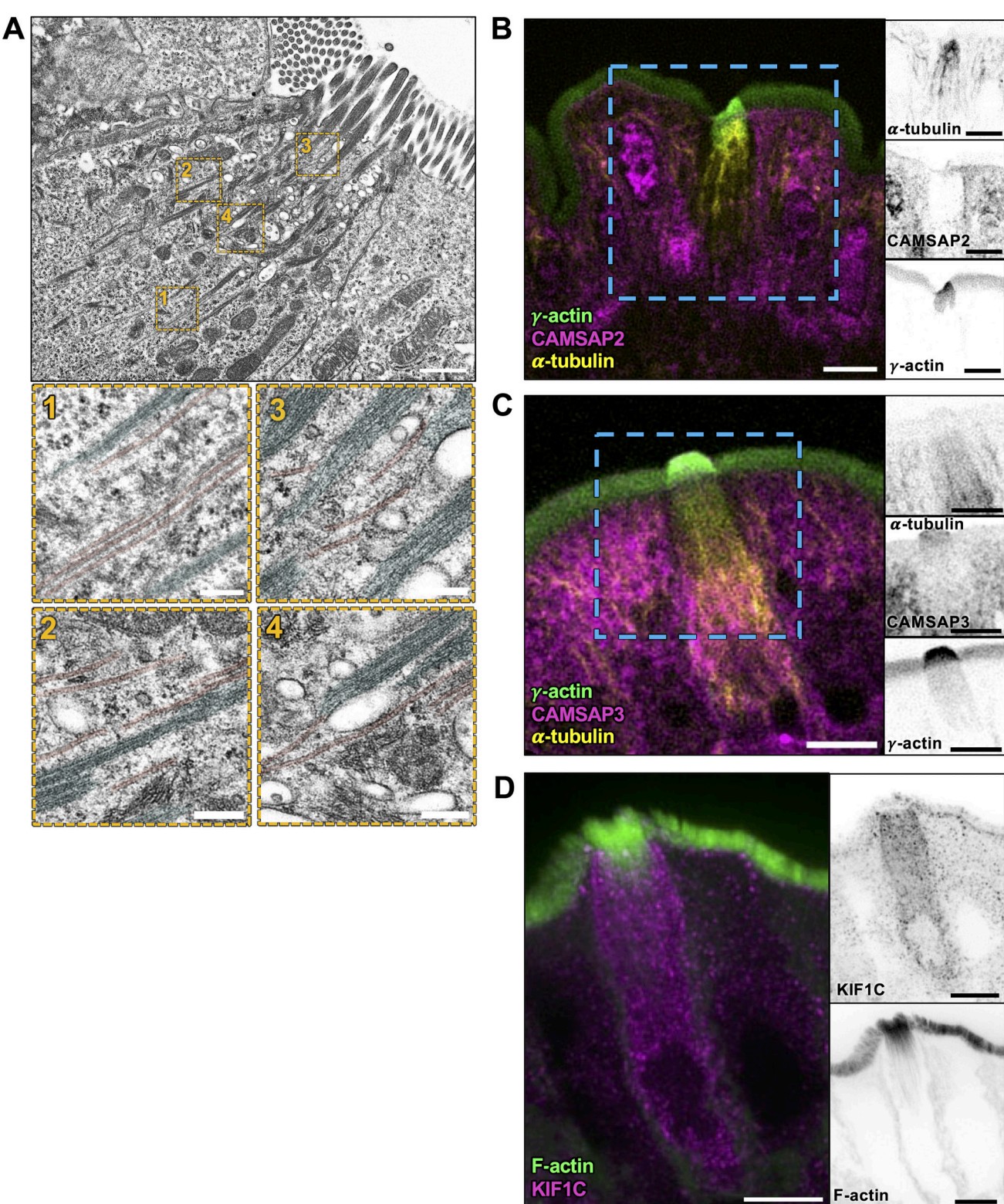

Figure S5. **Microtubules align with core actin bundles but lack clear polarity markers in tuft cells. (A)** TEM of ultrathin tissue slice showing lateral tuft cell section (scalebar = 1 µm) also shown in Fig. 7 A. Below, four zoomed areas show microtubules, pseudo-colored red, in proximity of the core actin bundle rootlets, pseudo-colored cyan (scalebar = 200 nm). **(B and C)** MaxIP SDC image of paraffin-embedded tissue and immunostained for (B) CAMSAP2 and (C) CAMSAP3 with microtubules marked by staining for α-tubulin and actin marked with staining for γ-actin (scalebar = 5 µm). **(D)** MaxIP SDC image of frozen tissue section immunostained for KIF1C with F-actin marked by phalloidin staining.

JCB

Video 1. **Tomographic EM data showing a volume of the apical domain of an individual tuft cell; field size is 4.2 × 3.3 μm. 20 frames/second.**

