## [Peer Review File · The Journal of Cell Biology]

Organization of a cytoskeletal superstructure in the apical domain of intestinal tuft cells

Jennifer Silverman, Evan Krystofiak, Leah Caplan, Ken Lau, and Matthew Tyska

Corresponding Author(s): Matthew Tyska, Vanderbilt University School of Medicine

Review Timeline:

Submission Date:	2024-04-13
Editorial Decision:	2024-05-28
Revision Received:	2024-08-26
Editorial Decision:	2024-09-04
Revision Received:	2024-09-13

Monitoring Editor: Greg Alushin

Scientific Editor: Dan Simon

Transaction Report:

DOI: <https://doi.org/10.1083/jcb.202404070>

May 28, 2024

Re: JCB manuscript #202404070

Dr. Matthew J Tyska
Vanderbilt University School of Medicine
Dept. of Cell and Developmental Biology
MCN T2209
1161 21st Avenue South
Nashville, TN 37232

Dear Dr. Tyska,

Thank you for submitting your manuscript entitled "Organization of a cytoskeletal superstructure in the apical domain of intestinal tuft cells." The manuscript was assessed by expert reviewers, whose comments are appended to this letter. We invite you to submit a revision if you can address the reviewers' key concerns, as outlined here.

You will see that the Reviewers are enthusiastic about your study and ask for a few additional experiments to strengthen the conclusions. We agree that validating the inverse microtubule orientation with new assays is essential. We also feel that Reviewer 2's point #3 that co-staining for additional crosslinkers, even if they have previously been studied (e.g. plastin), would be useful to contextualize the "unusual" localization patterns identified here. Pursuing cytoskeletal disruption studies seems to us to be beyond the scope of the current study, which is to provide a framework for future mechanistic experiments.

GENERAL GUIDELINES:

Text limits: Character count for an Article is < 40,000, not including spaces. Count includes title page, abstract, introduction, results, discussion, and acknowledgments. Count does not include materials and methods, figure legends, references, tables, or supplemental legends.

Figures: Articles may have up to 10 main text figures. Figures must be prepared according to the policies outlined in our Instructions to Authors, under Data Presentation, <https://jcb.rupress.org/site/misc/ifora.xhtml>. All figures in accepted manuscripts will be screened prior to publication.

Supplemental information: There are strict limits on the allowable amount of supplemental data. Articles may have up to 5 supplemental figures. Up to 10 supplemental videos or flash animations are allowed. A summary of all supplemental material should appear at the end of the Materials and methods section.

Please note that JCB now requires authors to submit Source Data used to generate figures containing gels and Western blots with all revised manuscripts. This Source Data consists of fully uncropped and unprocessed images for each gel/blot displayed in the main and supplemental figures. If your revised paper will include cropped gel and/or blot images, please be sure to provide one Source Data file for each figure that contains gels and/or blots along with your revised manuscript files. File names for Source Data figures should be alphanumeric without any spaces or special characters (i.e., SourceDataF#, where F# refers to the associated main figure number or SourceDataFS# for those associated with Supplementary figures). The lanes of the gels/blots should be labeled as they are in the associated figure, the place where cropping was applied should be marked (with a box), and molecular weight/size standards should be labeled wherever possible. Source Data files will be made available to reviewers during evaluation of revised manuscripts and, if your paper is eventually published in JCB, the files will be directly linked to specific figures in the published article.

The typical timeframe for revisions is three to four months. While most universities and institutes have reopened labs and allowed researchers to begin working at nearly pre-pandemic levels, we at JCB realize that the lingering effects of the COVID-

19 pandemic may still be impacting some aspects of your work, including the acquisition of equipment and reagents. Therefore, if you anticipate any difficulties in meeting this aforementioned revision time limit, please contact us and we can work with you to find an appropriate time frame for resubmission. Please note that papers are generally considered through only one revision cycle, so any revised manuscript will likely be either accepted or rejected.

Thank you for this interesting contribution to Journal of Cell Biology. You can contact us at the journal office with any questions at cellbio@rockefeller.edu.

Sincerely,

Greg Alushin, PhD
Monitoring Editor
Journal of Cell Biology

Dan Simon, PhD
Scientific Editor
Journal of Cell Biology

Reviewer #1 (Comments to the Authors (Required)):

This study reports detailed ultrastructural and molecular organization of the intestinal tuft cells. These cells represent a relatively minor subpopulation in the intestinal epithelium and a few other tissues. The main focus of the current work is on the cytoskeleton of tuft cells, especially on the actin cytoskeleton in the apical microvillus-like protrusions ("tuft"). Previous structural studies of tuft cells either addressed different questions or did not produce sufficiently detailed information. Here, the authors applied cutting-edge imaging and analytical techniques to provide detailed high-quality information about actin filament organization in the tuft and how these actin bundles interact with microtubules and cytoplasmic vesicles in the cytoplasm. Additionally, the authors obtained new insights into molecular composition of the actin filament bundles in the tuft. Although this study can be broadly categorized as descriptive, it is expected to be of high impact because the authors have accomplished essential groundwork for subsequent mechanistic and functional studies. I highly recommend it for publication, but have some comments, mostly minor, which if addressed can further strengthen this excellent study.

1. One relatively serious concern is about the conclusion that tuft cells have an inverse orientation of microtubules, as compared with typical epithelial cells. This conclusion is based on dynein staining (Fig. 6E), which is not the best marker of microtubule polarity. Also, dynein does not seem to be well-expressed in this cell type, as can be judged from its weak staining signal, although a gradient toward the basal surface is detectable. Given the counterintuitive result about microtubule polarity, stronger evidence on this point would be beneficial. The authors may consider staining for CAMSAP or EB1. Although the microtubules are mostly stable (acetylated), their plus ends may still be dynamic and contain EB1.

2. It will be helpful on the first mention to provide other popular synonyms for both cells (e.g. tuft cell = brush cell) and proteins (e.g. LIMA1 = EPLIN).

3. It seems that only 3 tuft cells were analyzed by TEM, which is quite on a low end. If it is too difficult to find more cells, minimally, authors need to provide information on whether the data from individual cells agree with each other. If so, it would justify pooling the data together.

4. Fig. 2C: the number of analyzed bundles is not specified.

5. Fig. 2D and G: Can the authors extract numbers from these data, such as fractions of pink bundles in each case?

6. Given that the study focuses on the tuft cell cytoskeleton and that the authors used TEM, can they comment on the presence and distribution of intermediate filaments?

7. The diameters of individual actin bundles in the tuft and the diameters of apical protrusions would be other useful parameters to add.

8. P. 6, l. 149: "Right at the apical surface, bundle area and cell area are similar (Figure 1H)". In fact, the images in Figure 1D

and G conflict with this quantification result, because they show ~1-2 um gaps between the actin core and the cell edge. Either these images were taken below the apical surface, which then needs to be clarified, or the conclusion is not correct. A potential caveat for the quantification in Fig. 1H, is that it seems to be more appropriate to perform pairwise comparisons of the area of the cell with the area of the actin core in the same cell, rather than to calculate the averages for each value across the cell population.

9. More explanations are needed for the scRNAseq methods and results (Figures S1E,G and S3). For methods, the authors need to give at least an outline of what they did for this study, as the original paper seems to give too much irrelevant information. For the figures, the green-blue scale is not defined: Intensity of what? What are the units? Also, what is the data outside of red and grey circles?

10. Figures S1F and H: They show "grey value" of what: scRNA or immunofluorescence? Fix the y-axis labels in S1H.

11. Figure 2H: Purple shading in zoomed panels seems too dark, making it difficult to see actin filaments.

12. P. 10, l. 270: "myosin-1b is a tension sensitive motor implicated in vesicle secretion [54]." For tension sensitivity, the proper reference is PMID: 18599791. Alternatively, a review on these myosins can be cited to include both points.

13. In Figures 5G, S1A and S1B, $p < 0.001$ is denoted by 4 asterisks, but such level of significance is typically denoted by 3 asterisks. The data in Figure 8B and 8D have the same $p = 0.0015$, but the graphs are labeled with 3 or 2 asterisks, respectively.

14. To provide additional illustration to the points on microtubules, the authors can label microtubules either on the existing TEM images or add new ones.

15. P. 11, ll.329-330: "These EVs were similar in dimensions and appearance to the smaller vesicles noted in the sub-apical cytoplasm". It is more proper to compare EVs to intraluminal vesicles within multivesicular bodies, rather than to vesicles in the cytoplasm, which cannot be delivered to outside. It would also be a good idea to quantify and compare the sizes of EVs and intraluminal vesicles.

16. P. 13, ll. 383-385: "tight packing could be the result of compressive forces applied by the junctional contractile ring of F-actin and NM2, which encircles the cluster of giant core actin bundles." This conclusion seems rather weak, because the bundle is not well seen in any images (fluorescence or EM), as well as because of comment #8. Ideally, this conclusion should be better illustrated, or the statement should be toned down.

17. P. 22, l. 525: "Briefly, CRISPR/Cas9 genome engineering methods were used to insert a flexible linker and mCherry coding sequence at the 3' end of the CDHR5 terminal coding exon." Strictly speaking, the sequence of the linker needs to be provided, as this is a new construct.

18. Typos: p. 21, ll. 514-515 ("20 Transfections were performed using Lipofectamine 2000 (Thermo Fischer #11668019) 21 according to the manufacturer's protocol"); p. 23, l. 576 ("imaging was conducted using a using a Nikon Ti2 inverted light microscope"); fig. 1F legend ("yellow" should be "green"); legend for fig. 3C ("below" should be "above"); legend for Fig. 6B ("measurements demonstrated in Fig 7A" - which?)

Reviewer #2 (Comments to the Authors (Required)):

The manuscript entitled "Organization of a cytoskeletal superstructure in the apical domain of intestinal tuft cells" by Silverman et al., provides a comprehensive description of the structure of actin filaments and microtubules in rare specialized epithelial tuft cells. Throughout this work the authors provide visually stunning high-resolution microscopy and detailed quantification of immuno-stained cells and tissues, highlighting the localization of several cytoskeleton-related proteins identified from RNA-seq data. The most interesting finding is the organization of a "giant" (at least compared to microvilli) actin-based microvilli-like structure that is embedded deeply within the cell but also protrudes several microns from the cell base. The study also details a co-alignment and interdigitation between actin and microtubules in these cells, suggesting tuft cells have the potential for unique mechanisms of cytoskeletal crosstalk. Overall, this work is likely of interest to cytoskeletal enthusiasts and cell biologists but requires a bit more detail/clarification.

I have three main concerns:

1) The naming of the "giant" structure. Compared to microvilli, the cytoskeletal organization of tuft cells does seem to be giant. The authors claim up to 10x the size of microvilli, but probably closer to 3-5x from the representative TEM images provided. There are also many examples of thick actin bundles that extend across longer distances in other systems including, plants where bundles may extend tens of microns depending on the cell type; *Limulus* sperm where classic work suggests 55-micron long actin filaments are associated with microtubules; or even some axons. The authors should consider renaming this structure or at least use this reviewer's confusion as an opportunity to discuss their work in the context of these mechanisms. Are the same factors at play in setting up such intricate structures across different systems?

2)The functional implications of the cytoskeletal features in tuft cells are not tested. Thus, the study could benefit from some sort of cytoskeletal manipulation and a larger discussion/more clarity about what experiments are possible in this system. Ideally seeing the effect of the loss of some/one of the organizational proteins on the actin structures could provide some missing mechanistic details. If the pharmacological disruption of actin and microtubules is possible in this system, this may strengthen the idea that the co-alignment of actin and MTs is a functional feature of the system. Similarly, did the authors observe any direct crosslinking between actin and MTs or have any idea what proteins may mediate such ordered structure between those cytoskeletal elements?

3)The authors used RNAseq data to choose what proteins to stain for throughout this work. There is reasonable discussion on fimbrin being the likely crosslinker (and several previous studies have shown this). Even though it is already published, this reviewer kept looking for such images in this work. In a similar vein, a villin family member is investigated - it would be valuable to also include staining for the prominent villin in microvilli, even just for comparison. The TEM images are stunning, and the order of the structure also makes this reviewer curious if other crosslinkers (even though not the top hits from RNAseq) are present in these cells. Fascin and alpha-actinin could both easily fit in the same space - are they there? At a minimum discussing these top hits with classic bundlers/crosslinkers would extend the audience of this work.

Minor:

1)What makes tuft cells one of the rarest? This statement is a little confusing in the abstract without the context from the introduction.

2)The web-based form messed up the coding for the micron symbol.

3)A little more description on the function of tuft cells (and what diseases their dysfunction are related to) may help the uninitiated.

RESPONSE TO REVIEWER COMMENTS

JCB manuscript #202404070

"Organization of a cytoskeletal superstructure in the apical domain of intestinal tuft cells."
Silverman et al.

We thank the Reviewers for taking the time to provide valuable and constructive feedback on our manuscript. Both Reviewers raised important points for improving the work and we address each comment below in a point-by-point manner. The revised manuscript now contains clarifications in the text, as well as new results and figure panels that collectively strengthen the paper. In our response below, Reviewer comments are shown in black arial font, while our responses are provided in indented blue arial font.

Reviewer #1 (Comments to the Authors)

This study reports detailed ultrastructural and molecular organization of the intestinal tuft cells. These cells represent a relatively minor subpopulation in the intestinal epithelium and a few other tissues. The main focus of the current work is on the cytoskeleton of tuft cells, especially on the actin cytoskeleton in the apical microvillus-like protrusions ("tuft"). Previous structural studies of tuft cells either addressed different questions or did not produce sufficiently detailed information. Here, the authors applied cutting-edge imaging and analytical techniques to provide detailed high-quality information about the actin filament organization in the tuft and how these actin bundles interact with microtubules and cytoplasmic vesicles in the cytoplasm. Additionally, the authors obtained new insights into molecular composition of the actin filament bundles in the tuft. Although this study can be broadly categorized as descriptive, it is expected to be of high impact because the authors have accomplished essential groundwork for subsequent mechanistic and functional studies. I highly recommend it for publication, but have some comments, mostly minor, which if addressed can further strengthen this excellent study.

We thank the Reviewer for their comments and also want to acknowledge their remarkable attention to detail. Our corrections and addition of new data in response to these comments have significantly strengthened this manuscript.

1. One relatively serious concern is about the conclusion that tuft cells have an inverse orientation of microtubules as compared with typical epithelial cells. This conclusion is based on dynein staining (Fig. 6E), which is not the best marker of microtubule polarity. Also, dynein does not seem to be well-expressed in this cell type, as can be judged from its weak staining signal., although a gradient toward the basal surface is detectable. The authors may consider staining for CAMSAP or EB1. Although the microtubules are mostly stable (acetylated), their plus ends may still be dynamic and contain EB1.

Based on the Reviewer's points, we agree that the argument for minus-end out polarity of microtubules in the tuft cell superstructure is weak based on the dynein staining alone. To alleviate this concern, we stained for five other factors that are expected to accumulate at microtubule ends, including CAMSAPs and EB1.

Staining for minus-end binder CAMSAP1 resulted in diffuse signal that was similar across enterocytes and tuft cells and difficult to interpret. However, staining for CAMSAP2 (Figure S5B) revealed localization that was comparable to our original dynein staining (Figure 6C-E), characterized by decreased intensity throughout the sub-apical cytoplasm, in the space occupied by the super-structure. CAMSAP3 staining

demonstrated decreased signal throughout the space occupied by the super-structure, but with unexpected additional enrichment in apical protrusions (Figure S5B). We also stained for EB1, as the Reviewer suggested, but those experiments also unexpectedly showed strong staining at the tips of apical protrusions. Because our ultrastructural studies indicate that tuft cell protrusions are devoid of microtubules, we are not quite sure how to explain the localization of CAMSAP3 and EB1 in these structures, although it might reflect one of the actin-associated functions of these molecules [1-3]. Finally, informed by our scRNAseq data, we stained for plus-end directed kinesin KIF1C, which exhibited punctate labeling throughout the core bundle region and cytoplasm, but overall higher levels relative to adjacent enterocytes (Figure S5D).

Even with some similarities in dynein and CAMSAP2 localization, the variation across these multiple staining patterns makes it difficult for us to offer a strong conclusion on microtubule polarity in the superstructure. Nevertheless, in the revised paper, we now include CAMSAP2, CAMSAP3, and KIF1C staining to offer the reader additional points of characterization on microtubule binding proteins in this unique cytoskeletal feature. Based on these new data, we changed the wording in the manuscript to highlight the clear differences in microtubule organization between tuft cells and enterocytes, and point out that defining microtubule polarity in this context will likely require a more advanced ultrastructural approach (see pg. 12).

2. It will be helpful on the first mention to provide other popular synonyms for both cells (e.g. tuft cell = brush cell) and proteins (e.g. LIMA1 = EPLIN).

We now include the other synonyms for tuft cells (brush cells) and LIMA1 (EPLIN) when they are first mentioned in the manuscript on pg. 4 and pg. 10, respectively.

3. It seems that only 3 tuft cells were analyzed by TEM, which is quite on a low end. If it is too difficult to find more cells, minimally, authors need to provide information on whether the data from individual cells agree with each other. If so, it would justify pooling the data together.

In the legend for Figure 2, we now provide median values per tuft cell for the number of filaments per bundle (Figure 2B), core bundle diameter (Figure 2C), filament packing angles (Figure 2E), and distance between filaments (Figure 2F).

4. Fig 2C: the number of analyzed bundles is not specified.

Thank you for pointing this out; we now specify the number of bundles in the figure legend (note: this panel is now located in Figure S2B).

5. Fig. 2D and G: Can the authors extract numbers from these data such as fractions of pink bundles in each case?

We now list the range of percentage hexagonally packed based on the fraction of pink bundles observed, in the narrative on pg.8.

6. Given that the study focuses on the tuft cell cytoskeleton and that the authors used TEM, can they comment on the presence and distribution of intermediate filaments?

Previous TEM and immunofluorescence studies in other epithelial tissues (lung, gall bladder, and stomach) revealed intermediate filaments extending through the sub-apical region of tuft cells, oriented parallel to actin bundles [4, 5]. Based on those studies, we

stained for both CK-18 and neurofilaments in intestinal tissues. CK-18 did not stain well under the conditions we used for our tissue preparation, and neurofilament labeling was much lower in tuft cells relative to neighboring enterocytes (see below). In the narrative on pg. 17, we now point out the prior work in other tissues and leave open the possibility that intermediate filaments may also be co-aligned with actin and microtubule polymers in the sub-apical superstructure.

7. The diameters of individual actin bundles in the tuft and the diameters of apical protrusions would be other useful parameters to add.

We agree that adding the diameters of individual actin bundles and apical protrusions would be helpful and added these values to the manuscript. New quantification of these parameters can be found in Figure 2C (actin bundle diameter, median of 106.1 nm) and Figure S2A (apical protrusion diameter, median of 152.9 nm)

8. P. 6, l. 149: "Right at the apical surface, bundle area and cell area are similar (Figure 1H)." In fact, the images in Figure 1D and G conflict with this quantification result, because they show ~1-2 μm gaps between the actin core and the cell edge. Either these images were taken below the apical surface, which then needs to be clarified, or the conclusion is not correct. A potential caveat for the quantification in Fig. 1H, is that it seems to be more appropriate to calculate the averages for each value across the cell population.

We apologize that the wording on this point was confusing. The image in Figure 1H was acquired 3 μm below the apical surface. In the revised manuscript, we clarified this in the figure by including an additional text label on this panel, which shows where in the cell volume these planes were sampled.

9. More explanations are needed for the scRNAseq methods and results (Figures S1E, G and S3). For methods, the authors need to give at least an outline of what they did for this study, as the original paper seems to give too much irrelevant information. For the figures, the green-blue scale is not defined: Intensity of what? What are the units? Also, what is the data outside of the red and gray circles.

We now include a more detailed overview of the scRNA-seq workflow in the revised Methods section on pg. 26. We also added a 'key' to Figures S1E and S3A, which defines the cell types associated with each cluster on the plot, and the heatmap overlay, which represents an Arcsinh-scaled normalized transcript count.

10. Figures S1F and H: They show “gray value” of what: scRNA or immunofluorescence? Fix the y-axis labels in S1H.

Thank you for catching this; we fixed the axis labels in Figures S1F and S1H to clarify what they mean.

11. Figure 2H: Purple shading in zoomed panels seems too dark, making it difficult to see actin filaments.

To improve visualization, we switched the highlighting from a purple fill to a green outline.

12. P. 10, l. 270: “myosin-1b is a tension sensitive motor implicated in vesicle secretion [54].” For tension sensitivity, the proper reference is PMID: 18599791. Alternatively, a review on these myosins can be cited to include both points.

Thank you for your attention to detail and catching this mistake. We corrected the reference to reflect the proper study.

13. In Figures 5G, S1A and S1B, $p < 0.001$ is denoted by 4 asterisks, but such level of significance is typically denoted by 3 asterisks. The data in Figure 8B and 8D have the same $p = 0.0015$, but the graphs are labeled with 3 or 2 asterisks, respectively.

We corrected this issue in Figures 5G and S1B. The significance reported in 8B was a typo and was fixed to report the true significance of $p = 0.0005$, corresponding to three asterisks.

14. To provide additional illustration to the points on microtubules, the authors can label microtubules either on the existing TEM images or add new ones.

This is a great suggestion, and we now include a gallery of microtubule images from TEM images of this sub-apical region in Figure S5A. In these EM images, microtubules are readily identified running parallel to actin core bundles, as individual polymers and in bundles. To provide another ultrastructural perspective, we also include a new animation of a tomographic EM volume (Video S1), which also shows microtubules running between and parallel to actin core bundles. Additionally, this tomogram clearly shows that the core bundles extend continuously from the apical protrusions down into the sub-apical cytoplasm, corroborating the conclusions made on pg. 6 of the manuscript.

15. P 11, ll.329-330: “These EVs were similar in dimensions and appearances to the smaller vesicles noted in the sub-apical cytoplasm”. It is more proper to compare EVs to intraluminal vesicles within multivesicular bodies, rather than to vesicles in the cytoplasm, which cannot be delivered to the outside. It would also be a good idea to quantify and compare the sizes of EVs and intraluminal vesicles.

We agree that the small vesicles we compared to EVs are better described as 'intraluminal vesicles within multivesicular bodies', and we edited the text to reflect this point. Additionally, we quantified the diameter of the EVs (44 ± 23 nm) and intraluminal vesicles (45 ± 14 nm) and now include these values in the text on pg. 13.

16. P. 13, ll.383-385: "tight packing could be the result of compressive forces applied by the junctional contractile ring of F-actin and NM2, which encircles the cluster of giant core actin bundles". This conclusion seems rather weak, because the bundle is not well seen in any images (fluorescence or EM), as well as because of comment #8. Ideally, this conclusion should be better illustrated, or the statement should be toned down.

We agree with the Reviewer that the conclusion was weak and have removed this sentence from the manuscript.

17. P. 22, l. 525: "Briefly, CRISPR/Cas9 genome engineering methods were used to insert a flexible linker and mCherry coding sequence at the 3' end of the CDHR5 terminal coding exon". Strictly speaking, the sequence of the linker needs to be provided, as this is a new construct.

We now provide the linker sequence for the CDHR5-mCherry mouse in the Methods on pg. 24.

18. Typos: p. 21, ll. 514-515 ("20 transfections were performed using Lipofectamine 2000 (Thermo Fisher #11668019) 21 according to the manufacture's protocol"); p.23, l.576 ("imaging was conducted using a using a Nikon Ti2 inverted light microscope"); fig. 1F legend ("yellow" should be "green"); legend for fig. 3C ("below" should be "above"); legend for Fig. 6B ("Measurements demonstrated in Fig 7A" – which?)

Thank you for finding these typos; all of them were corrected in the revised manuscript.

Reviewer #2 (Comments to the Authors):

The manuscript entitled "Organization of a cytoskeletal superstructure in the apical domain of intestinal tuft cells" by Silverman et al., provides a comprehensive description of the structure of the actin filaments and microtubules in the rare specialized epithelial tuft cells. Throughout this work the authors provide visually stunning high-resolution microscopy and detailed quantification of the immune-stained cells and tissues, highlighting the localization of several cytoskeleton-related proteins identified from RNA-seq data. The most interesting finding is the organization of a "giant" (at least compared to microvilli) actin-based microvilli-like structure that is embedded deeply within the cell but also protrudes several microns from the cell base. The study also details a co-alignment and interdigitation between actin and microtubules in these cells, suggesting tuft cells have the potential for unique mechanisms of cytoskeletal crosstalk. Overall, this work is likely of interest to cytoskeletal enthusiasts and cell biologists but requires a bit more detail/clarification.

We thank this Reviewer for taking their time to review our work, and for their insightful comments. The details and suggestions raised below helped us focus our efforts during the revision process as we worked to strengthen the manuscript.

I have three main concerns:

1a. The naming of the “giant” structure. Compared to microvilli, the cytoskeletal organization of tuft cells does seem to be giant. The authors claim up to 10x the size of microvilli but probably closer to 3-5x from the representative TEM images provided.

We agree that ‘giant’ might not have been the best label here, especially considering the other cytoskeletal examples cited by the Reviewer. Throughout the revised manuscript, we changed our phrasing to reflect more conventional language; giant actin bundles and giant rootlets are now referred to as core bundles and rootlets, respectively. Because tuft cell protrusions are clearly microvillus-like, we still draw comparisons to the much smaller microvilli found in the enterocyte brush border, when discussing the dimensions of these structures (‘microvillus-like’ added on pg. 6).

1b. There are also many examples of thick actin bundles that extend across longer distances in other systems including, plants where bundles may extend tens of microns depending on the cell type;...

The actin cables found in *Nitella* and *Chara* are great examples of polarized actin bundles that extend extremely long distances (up to hundreds of μm), and we now mention these in the narrative on pg. 15.

1c. *Limulus* sperm where classic work suggests 55-micron long actin filaments are associated with microtubules; or even some axons. The authors should consider renaming this structure or at least using this reviewer’s confusion as an opportunity to discuss their work in the context of these mechanisms. Are the same factors at play in setting up such intricate structures across different systems?

The Reviewer’s comment here suggests that microtubules associate with the $>50 \mu\text{m}$ long actin bundles that form during the acrosomal reaction in *Limulus*. However, we were unable to find clear data to support this point (if we missed it, we would welcome a correction). Classic TEM images do show that these bundles make passing contact with the basal body that supports the flagellum in these sperm cells (see Figure 2, [6]), but that would be entirely different from the architecture we are describing here for tuft cells, where large numbers of actin bundles and microtubules exhibit interdigitation in the sub-apical cytoplasm. Nevertheless, in the Discussion on pg. 15, we now refer to the other striking examples of exaggerated cytoskeletal features, as pointed out by the Reviewer here and above. We also allude to the overlap of actin filaments and microtubules that occurs in the context of the neuronal axons on pg. 17.

2. The functional implications of the cytoskeletal features in tuft cells are not tested. Thus, the study could benefit from some sort of cytoskeletal manipulation and a larger discussion/more clarity about what experiments are possible in this system. Ideally seeing the effect of the loss of some/one of the organizational proteins on the actin structures could provide some missing mechanistic details. If the pharmacological disruption of actin and microtubules is possible in this system, this may strengthen the idea that co-alignment of actin and MTs is a functional feature of the system. Similarly, did the authors observe any direct crosslinking between actin and MTs or have any idea what proteins may mediate such ordered structure between those cytoskeletal systems.

We agree with this reviewer; loss-of-function studies are an essential next step toward defining the cell biological mechanisms that lead to the unique morphology and function of the tuft cell. We probably could have stated this more clearly in the original

manuscript, but ***the main obstacle to carrying out such experiments is the lack of an intestinal epithelial cell culture model that produces tuft cells*** (we now make this clear in the manuscript on pg. 5). Currently, we are limited to mouse models for conducting perturbation studies of this type and, given the extended timeline for building and characterizing those animals (and the practical matter of having little room for adding new data to the manuscript in its current form), we believe those studies rest well outside the scope of the current paper.

3. The authors used RNAseq data to choose what proteins to stain for throughout this work. There is reasonable discussion on fimbrin being the likely crosslinker (and several previous studies have shown this). Even though it is already published, this reviewer kept looking for such images in this work. In a similar vein, a villin family member is investigated – it would be valuable to also include staining for the prominent villin in microvilli, even just for comparison. The TEM images are stunning, and the order of the structure also makes this reviewer curious if other crosslinkers (even though not top hits from RNAseq) are present in these cells. Fascin and alpha-actinin could both easily fit in the same space – are they there? At a minimum discussing these top hits with classic bundlers/crosslinkers would extend the audience of this work.

We agree that including actin-binding proteins that were previously identified in tuft cells would benefit the manuscript. Because previous immunofluorescence studies confirmed the presence of fimbrin and phosphorylated girdin Y1798 (pGirdin) in tuft cells, we first revisited the localization of these factors using super-resolution Airyscan imaging. We now include Airyscan images and linescan quantifications for fimbrin and pGirdin in Figure 4 and refer to this data in the manuscript on pg. 10.

Villin, the major actin-bundling protein in enterocyte microvilli, was also allegedly found in tuft cells as reported in early studies [7]. However, later studies indicated that structurally related advillin, rather than villin, is most highly enriched in tuft cells [8, 9].

We also stained for fascin and α -actinin based on the Reviewer's suggestions. While we observed fascin in the apical tuft at slightly higher levels than the surrounding structures (see below), we were not comfortable enough with the quality of the staining to pursue quantification of its localization. However, we did observe that α -actinin-4 enriched in tuft cell protrusions, and we have added new panels with this data to Figure S4 and refer to this point in the manuscript on pg. 10.

Related to this point, while our paper was in peer review, we also probed tuft cells for candidate motors that might use the cytoskeletal superstructure as a track to drive cargo transport. Here we stained for myosin-6, myosin-5b and myosin-7b, all of which are present in enterocytes [10-12]. Interestingly, we found strong signals for all

three of these myosins in tuft cells. Myosin-6, the minus-end directed motor tied to endocytosis, was clearly enriched in tuft cells relative to enterocytes (Figure 8E, F). Myosin-5b, which is linked to vesicle recycling, was enriched at the base of tuft cell protrusions at levels comparable to enterocytes (Figure 8G, H). Myosin-7b, a component of the intermicrovillar adhesion complex, was found at the distal tips of tuft protrusions, at much higher levels than surrounding enterocytes (Figure S8 I, J). These new findings were incorporated into the manuscript on pgs. 13,14, & 17.

Minor:

1. What makes tuft cells one of the rarest? This statement is a little confusing in the abstract without the context from the introduction.

Tuft cells generally make up around 1-10% of the cell population based on the tissue; the small intestinal epithelium is composed of ~1% of tuft cells at homeostasis. While this cell type is more rare than other absorptive cells, we realize the current phrasing may be too specific. Therefore, in the abstract we changed the phrasing from 'one of the rarest cell types...' to 'Tuft cells are a rare epithelial cell type...'

2. The web-based form messed up the coding on the micron symbol

Our apologies - we hope we avoided this problem with our resubmission.

3. A little more description on the function of tuft cells (and what diseases their dysfunction are related to) may help the uninitiated.

To address this point, we added additional clarifying information on the importance of tuft cells in parasite clearance. Here we highlighted a landmark study from **Gerbe et al. [13]** et al. where tuft cells were eliminated in mice, which were then challenged with a helminth infection. Unlike wild-type mice, tuft cell-deficient mice were unable to clear the worms, with the infection spreading through additional parts of the intestine. These edits can be found on pg. 4 of the revised manuscript.

REFERENCES

1. Ning, W., et al., *The CAMSAP3-ACF7 Complex Couples Noncentrosomal Microtubules with Actin Filaments to Coordinate Their Dynamics*. Dev Cell, 2016. **39**(1): p. 61-74.
2. Noordstra, I., et al., *Control of apico-basal epithelial polarity by the microtubule minus-end-binding protein CAMSAP3 and spectraplakins ACF7*. J Cell Sci, 2016. **129**(22): p. 4278-4288.
3. Alberico, E.O., et al., *Interactions between the Microtubule Binding Protein EB1 and F-Actin*. J Mol Biol, 2016. **428**(6): p. 1304-1314.
4. Luciano, L., S. Groos, and E. Reale, *Brush cells of rodent gallbladder and stomach epithelia express neurofilaments*. J Histochem Cytochem, 2003. **51**(2): p. 187-98.
5. Kasper, M., et al., *Colocalization of cytokeratin 18 and villin in type III alveolar cells (brush cells) of the rat lung*. Histochemistry, 1994. **101**(1): p. 57-62.
6. Tilney, L.G., *Actin filaments in the acrosomal reaction of Limulus sperm. Motion generated by alterations in the packing of the filaments*. J Cell Biol, 1975. **64**(2): p. 289-310.

7. Hofer, D. and D. Drenckhahn, *Identification of brush cells in the alimentary and respiratory system by antibodies to villin and fimbrin*. Histochemistry, 1992. **98**(4): p. 237-42.
8. Esmailniakooshkghazi, A., et al., *Mouse intestinal tuft cells express advillin but not villin*. Sci Rep, 2020. **10**(1): p. 8877.
9. Ruppert, A.L., et al., *Advillin is a tuft cell marker in the mouse alimentary tract*. J Mol Histol, 2020. **51**(4): p. 421-435.
10. Weck, M.L., et al., *Myosin-7b Promotes Distal Tip Localization of the Intermicrovillar Adhesion Complex*. Curr Biol, 2016. **26**(20): p. 2717-2728.
11. Lapierre, L.A., et al., *Myosin vb is associated with plasma membrane recycling systems*. Mol Biol Cell, 2001. **12**(6): p. 1843-57.
12. Buss, F., et al., *Myosin VI isoform localized to clathrin-coated vesicles with a role in clathrin-mediated endocytosis*. EMBO J, 2001. **20**(14): p. 3676-84.
13. Gerbe, F., et al., *Intestinal epithelial tuft cells initiate type 2 mucosal immunity to helminth parasites*. Nature, 2016. **529**(7585): p. 226-30.

September 4, 2024

RE: JCB Manuscript #202404070R

Dr. Matthew J Tyska
Vanderbilt University School of Medicine
Dept. of Cell and Developmental Biology
MCN T2209
1161 21st Avenue South
Nashville, TN 37232

Dear Dr. Tyska,

Thank you for submitting your revised manuscript entitled "Organization of a cytoskeletal superstructure in the apical domain of intestinal tuft cells." We would be happy to publish your paper in JCB pending final revisions necessary to meet our formatting guidelines (see details below).

A. MANUSCRIPT ORGANIZATION AND FORMATTING:

1) Text limits: Character count for Articles is < 40,000, not including spaces. Count includes title page, abstract, introduction, results, discussion, and acknowledgments. Count does not include materials and methods, figure legends, references, tables, or supplemental legends.

2) Figure formatting: Articles may have up to 10 main text figures. Scale bars must be present on all microscopy images, including inset magnifications. Please add scale bars to Figures 1K (inset magnification), 3C, & 6E.

Also, please avoid pairing red and green for images and graphs to ensure legibility for color-blind readers. If red and green are paired for images, please ensure that the particular red and green hues used in micrographs are distinctive with any of the colorblind types. If not, please modify colors accordingly or provide separate images of the individual channels.

3) Statistical analysis: Error bars on graphic representations of numerical data must be clearly described in the figure legend. The number of independent data points (n) represented in a graph must be indicated in the legend. Please, indicate whether 'n' refers to technical or biological replicates (i.e. number of analyzed cells, samples or animals, number of independent experiments). If independent experiments with multiple biological replicates have been performed, we recommend using distribution-reproducibility SuperPlots (please see Lord et al., JCB 2020) to better display the distribution of the entire dataset, and report statistics (such as means, error bars, and P values) that address the reproducibility of the findings.

Statistical methods should be explained in full in the materials and methods. For figures presenting pooled data the statistical measure should be defined in the figure legends. Please also be sure to indicate the statistical tests used in each of your experiments (both in the figure legend itself and in a separate methods section) as well as the parameters of the test (for example, if you ran a t-test, please indicate if it was one- or two-sided, etc.). Also, if you used parametric tests, please indicate if the data distribution was tested for normality (and if so, how). If not, you must state something to the effect that "Data distribution was assumed to be normal but this was not formally tested."

4) Materials and methods: Should be comprehensive and not simply reference a previous publication for details on how an experiment was performed. Please provide full descriptions (at least in brief) in the text for readers who may not have access to referenced manuscripts. The text should not refer to methods "...as previously described."

5) For all cell lines, vectors, constructs/cDNAs, etc. - all genetic material: please include database / vendor ID (e.g., Addgene, ATCC, etc.) or if unavailable, please briefly describe their basic genetic features, even if described in other published work or gifted to you by other investigators (and provide references where appropriate). Please be sure to provide the sequences for all of your oligos: primers, si/shRNA, RNAi, gRNAs, etc. in the materials and methods. You must also indicate in the methods the source, species, and catalog numbers/vendor identifiers (where appropriate) for all of your antibodies, including secondary. If antibodies are not commercial, please add a reference citation if possible.

6) Microscope image acquisition: The following information must be provided about the acquisition and processing of images:

- a. Make and model of microscope
- b. Type, magnification, and numerical aperture of the objective lenses
- c. Temperature
- d. Imaging medium
- e. Fluorochromes
- f. Camera make and model
- g. Acquisition software
- h. Any software used for image processing subsequent to data acquisition. Please include details and types of operations involved (e.g., type of deconvolution, 3D reconstitutions, surface or volume rendering, gamma adjustments, etc.).

7) References: There is no limit to the number of references cited in a manuscript. References should be cited parenthetically in the text by author and year of publication. Abbreviate the names of journals according to PubMed.

8) Supplemental materials: Articles may have up to 5 supplemental figures and 10 videos.

Please also note that tables, like figures, should be provided as individual, editable files. A summary of all supplemental material should appear at the end of the Materials and methods section. Please include one brief sentence per item.

9) Video legends: Should describe what is being shown, the cell type or tissue being viewed (including relevant cell treatments, concentration and duration, or transfection), the imaging method (e.g., time-lapse epifluorescence microscopy), what each color represents, how often frames were collected, the frames/second display rate, and the number of any figure that has related video stills or images.

10) eTOC summary: A ~40-50 word summary that describes the context and significance of the findings for a general readership should be included on the title page. The statement should be written in the present tense and refer to the work in the third person. It should begin with "First author name(s) et al..." to match our preferred style.

11) Conflict of interest statement: JCB requires inclusion of a statement in the acknowledgements regarding competing financial interests. If no competing financial interests exist, please include the following statement: "The authors declare no competing financial interests." If competing interests are declared, please follow your statement of these competing interests with the following statement: "The authors declare no further competing financial interests."

12) A separate author contribution section is required following the Acknowledgments in all research manuscripts. All authors should be mentioned and designated by their first and middle initials and full surnames. We encourage use of the CRediT nomenclature (<https://casrai.org/credit/>).

13) ORCID IDs: ORCID IDs are unique identifiers allowing researchers to create a record of their various scholarly contributions in a single place. Please note that ORCID IDs are required for all authors. At resubmission of your final files, please be sure to provide your ORCID ID and those of all co-authors.

14) Journal of Cell Biology now requires a data availability statement for all research article submissions. These statements will be published in the article directly above the Acknowledgments. The statement should address all data underlying the research presented in the manuscript. Please visit the JCB instructions for authors for guidelines and examples of statements at (<https://rupress.org/jcb/pages/editorial-policies#data-availability-statement>).

B. FINAL FILES:

**The license to publish form must be signed before your manuscript can be sent to production. A link to the electronic license to publish form will be sent to the corresponding author only. Please take a moment to check your funder requirements before

choosing the appropriate license.**

Thank you for your attention to these final processing requirements. Please revise and format the manuscript and upload materials within 7 days. If you need an extension for whatever reason, please let us know and we can work with you to determine a suitable revision period.

Thank you for this interesting contribution, we look forward to publishing your paper in Journal of Cell Biology.

Sincerely,

Greg Alushin, PhD
Monitoring Editor
Journal of Cell Biology

Dan Simon, PhD
Scientific Editor
Journal of Cell Biology